# Revisiting Plasticity in Visual Reinforcement Learning: Data, Modules and Training Stages

**Guozheng Ma**[1*]   **Lu Li**[1*]   **Sen Zhang**[2]   **Zixuan Liu**[1]   **Zhen Wang**[2]
**Yixin Chen**[3]   **Li Shen**[4†]   **Xueqian Wang**[1†]   **Dacheng Tao**[5]

[1]Tsinghua University   [2]The University of Sydney   [3]Washington University in St. Louis
[4]JD Explore Academy   [5]Nanyang Technological University, Singapore

{mgz21,lilu21,zx-liu21}@mails.tsinghua.edu.cn; sen.zhang@sydney.edu.au
zwan4121@uni.sydney.edu.au; chen@cse.wustl.edu; mathshenli@gmail.com
wang.xq@sz.tsinghua.edu.cn; dacheng.tao@ntu.edu.sg

## Abstract

Plasticity, the ability of a neural network to evolve with new data, is crucial for high-performance and sample-efficient visual reinforcement learning (VRL). Although methods like resetting and regularization can potentially mitigate plasticity loss, the influences of various components within the VRL framework on the agent's plasticity are still poorly understood. In this work, we conduct a systematic empirical exploration focusing on three primary underexplored facets and derive the following insightful conclusions: (1) data augmentation is essential in maintaining plasticity; (2) the critic's plasticity loss serves as the principal bottleneck impeding efficient training; and (3) without timely intervention to recover critic's plasticity in the early stages, its loss becomes catastrophic. These insights suggest a novel strategy to address the high replay ratio (RR) dilemma, where exacerbated plasticity loss hinders the potential improvements of sample efficiency brought by increased reuse frequency. Rather than setting a static RR for the entire training process, we propose *Adaptive RR*, which dynamically adjusts the RR based on the critic's plasticity level. Extensive evaluations indicate that *Adaptive RR* not only avoids catastrophic plasticity loss in the early stages but also benefits from more frequent reuse in later phases, resulting in superior sample efficiency.

## 1 Introduction

The potent capabilities of deep neural networks have driven the brilliant triumph of deep reinforcement learning (DRL) across diverse domains (Silver et al., 2017; Jumper et al., 2021; OpenAI, 2023). Nevertheless, recent studies highlight a pronounced limitation of neural networks: they struggle to maintain adaptability and learning from new data after training on a non-stationary objective (Lyle et al., 2021; Sokar et al., 2023), a challenge known as **plasticity loss** (Lyle et al., 2023; Abbas et al., 2023). Since RL agents must continuously adapt their policies through interacting with environment, non-stationary data streams and optimization objectives are inherently embedded within the DRL paradigm (Kumar et al., 2023). Consequently, plasticity loss presents a fundamental challenge for achieving sample-efficient DRL applications (Nikishin et al., 2022; D'Oro et al., 2022).

Although several strategies have been proposed to address this concern, previous studies primarily focused on mitigating plasticity loss through methods such as resetting the parameters of neurons (Nikishin et al., 2022; D'Oro et al., 2022; Sokar et al., 2023), incorporating regularization techniques (Kumar et al., 2023; Lyle et al., 2023; 2021) and adjusting network architecture (Schwarzer et al., 2023; Dohare et al., 2021; Abbas et al., 2023). The nuanced impacts of various dimensions within the DRL framework on plasticity remain underexplored. This knowledge gap hinders more precise interventions to better preserve plasticity. To this end, this paper delves into the nuanced mechanisms underlying DRL's plasticity loss from three primary yet underexplored perspectives: data, agent modules, and training stages. Our investigations focus on visual RL (VRL) tasks that enable decision-making directly from high-dimensional observations. As a representative paradigm

---

*Equal Contribution, †Corresponding authors.

of end-to-end DRL, VRL is inherently more challenging than learning from handcrafted state inputs, leading to its notorious sample inefficiency (Ma et al., 2022; Yarats et al., 2021a; Tomar et al., 2022).

We begin by revealing the indispensable role of data augmentation (DA) in mitigating plasticity loss for off-policy VRL algorithms. Although DA is extensively employed to enhance VRL's sample efficiency (Yarats et al., 2020; 2021a), its foundational mechanism is still largely elusive. Our investigation employs a factorial experiment with DA and Reset. The latter refers to the re-initialization of subsets of neurons and has been shown to be a direct and effective method for mitigating plasticity loss (Nikishin et al., 2022). However, our investigation has surprisingly revealed two notable findings: (1) Reset can significantly enhance performance in the absence of DA, but show limited or even negative effects when DA is applied. This suggests a significant plasticity loss when DA is not employed, contrasted with minimal or no plasticity loss when DA is utilized. (2) Performance with DA alone surpasses that of reset or other interventions without employing DA, highlighting the pivotal role of DA in mitigating plasticity loss. Furthermore, the pronounced difference in plasticity due to DA's presence or absence provides compelling cases for comparison, allowing a deeper investigation into the differences and developments of plasticity across different modules and stages.

We then dissect VRL agents into three core modules: the encoder, actor, and critic, aiming to identify which components suffer most from plasticity loss and contribute to the sample inefficiency of VRL training. Previous studies commonly attribute the inefficiency of VRL training to the challenges of constructing a compact representation from high-dimensional observations (Tomar et al., 2022; Laskin et al., 2020a; Wang et al., 2022; Stooke et al., 2021; Shah & Kumar, 2021). A natural corollary to this would be the rapid loss of plasticity in the encoder when learning from scratch solely based on reward signals, leading to sample inefficiency. However, our comprehensive experiments reveal that it is, in fact, the plasticity loss of the critic module that presents the critical bottleneck for training. This insight aligns with recent empirical studies showing that efforts to enhance the representation of VRL agents, such as meticulously crafting self-supervised learning tasks and pre-training encoders with extra data, fail to achieve higher sample efficiency than simply applying DA alone (Li et al., 2022b; Hansen et al., 2023). Tailored interventions to maintain the plasticity of critic module provide a promising path for achieving sample-efficient VRL in future studies.

Given the strong correlation between the critic's plasticity and training efficiency, we note that the primary contribution of DA lies in facilitating the early-stage recovery of plasticity within the critic module. Subsequently, we conduct a comparative experiment by turning on or turning off DA at certain training steps and obtain two insightful findings: (1) Once the critic's plasticity has been recovered to an adequate level in the early stage, there's no need for specific interventions to maintain it. (2) Without timely intervention in the early stage, the critic's plasticity loss becomes catastrophic and irrecoverable. These findings underscore the importance of preserving critic plasticity during the initial phases of training. Conversely, plasticity loss in the later stages is not a critical concern. To conclude, the main takeaways from our revisiting can be summarized as follows:

- DA is indispensable for preserving the plasticity of VRL agents. (Section 3)
- Critic's plasticity loss is a critical bottleneck affecting the training efficiency. (Section 4)
- Maintaining plasticity in the early stages is crucial to prevent irrecoverable loss. (Section 5)

We conclude by addressing a longstanding question in VRL: how to determine the appropriate replay ratio (RR), defined as the number of gradient updates per environment step, to achieve optimal sample efficiency (Fedus et al., 2020). Prior research set a static RR for the entire training process, facing a dilemma: while increasing the RR of off-policy algorithms should enhance sample efficiency, this improvement is offset by the exacerbated plasticity loss (Nikishin et al., 2022; Sokar et al., 2023; Schwarzer et al., 2023). However, aforementioned analysis indicates that the impact of plasticity loss varies throughout training stages, advocating for an adaptive adjustment of RR based on the stage, rather than setting a static value. Concurrently, the critic's plasticity has been identified as the primary factor affecting sample efficiency, suggesting its level as a criterion for RR adjustment. Drawing upon these insights, we introduce a simple and effective method termed *Adaptive RR* that dynamically adjusts the RR according to the critic's plasticity level. Specifically, *Adaptive RR* commences with a lower RR during the initial training phases and elevates it upon observing significant recovery in the critic's plasticity. Through this approach, we effectively harness the sample efficiency benefits of a high RR, while skillfully circumventing the detrimental effects of escalated plasticity loss. Our comprehensive evaluations on the DeepMind Control suite (Tassa et al., 2018) demonstrate that *Adaptive RR* attains superior sample efficiency compared to static RR baselines.

## 2 RELATED WORK

In this section, we briefly review prior research works on identifying and mitigating the issue of plasticity loss, as well as on the high RR dilemma that persistently plagues off-policy RL algorithms for more efficient applications. Further discussions on related studies can be found in Appendix A.

**Plasticity Loss.** Recent studies have increasingly highlighted a major limitation in neural networks where their learning capabilities suffer catastrophic degradation after training on non-stationary objectives (Sokar et al., 2023; Nikishin et al., 2024). Different from supervised learning, the non-stationarity of data streams and optimization objectives is inherent in the RL paradigm, necessitating the confrontation of this issues, which has been recognized by several terms, including primacy bias (Nikishin et al., 2022), dormant neuron phenomenon (Sokar et al., 2023), implicit under-parameterization (Kumar et al., 2020), capacity loss (Lyle et al., 2021), and more broadly, plasticity loss (Lyle et al., 2023; Kumar et al., 2023). Agents lacking plasticity struggle to learn from new experiences, leading to extreme sample inefficiency or even entirely ineffective training.

The most straightforward strategy to tackle this problem is to re-initialize a part of the network to regain rejuvenated plasticity (Nikishin et al., 2022; D'Oro et al., 2022; Schwarzer et al., 2023). However, periodic *Reset* (Nikishin et al., 2022) may cause sudden performance drops, impacting exploration and requiring extensive gradient updates to recover. To circumvent this drawback, *ReDo* (Sokar et al., 2023) selectively resets the dormant neurons, while *Plasticity Injection* (Nikishin et al., 2024) introduces a new initialized network for learning and freezes the current one as residual blocks. Another line of research emphasizes incorporating explicit regularization or altering the network architecture to mitigate plasticity loss. For example, Kumar et al. (2023) introduces *L2-Init* to regularize the network's weights back to their initial parameters, while Abbas et al. (2023) employs *Concatenated ReLU* (Shang et al., 2016) to guarantee a non-zero gradient. Although existing methods have made progress in mitigating plasticity loss, the intricate effects of various dimensions in the DRL framework on plasticity are poorly understood. In this paper, we aim to further explore the roles of data, modules, and training stages to provide a comprehensive insight into plasticity.

**High RR Dilemma.** Experience replay, central to off-policy DRL algorithms, greatly improves the sample efficiency by allowing multiple reuses of data for training rather than immediate discarding after collection (Fedus et al., 2020). Given the trial-and-error nature of DRL, agents alternate between interacting with the environment to collect new experiences and updating parameters based on transitions sampled from the replay buffer. The number of agent updates per environment step is usually called replay ratio (RR) (Fedus et al., 2020; D'Oro et al., 2022) or update-to-data (UTD) ratio (Chen et al., 2020; Smith et al., 2022). While it's intuitive to increase the RR as a strategy to improve sample efficiency, doing so naively can lead to adverse effects (Li et al., 2022a; Lyu et al., 2024). Recent studies have increasingly recognized plasticity loss as the primary culprit behind the high RR dilemma (Sokar et al., 2023; Nikishin et al., 2022; 2024). Within a non-stationary objective, an increased update frequency results in more severe plasticity loss. Currently, the most effective method to tackle this dilemma is to continually reset the agent's parameters when setting a high RR value (D'Oro et al., 2022). Our investigation offers a novel perspective on addressing this long-standing issue. Firstly, we identify that the impact of plasticity loss varies across training stages, implying a need for dynamic RR adjustment. Concurrently, we determine the critic's plasticity as crucial for model capability, proposing its level as a basis for RR adjustment. Drawing from these insights, we introduce *Adaptive RR*, a universal method that both mitigates early catastrophic plasticity loss and harnesses the potential of high RR in improving sample efficiency.

## 3 DATA: DATA AUGMENTATION IS ESSENTIAL IN MAINTAINING PLASTICITY

In this section, we conduct a factorial analysis of DA and Reset, illustrating that DA effectively maintains plasticity. Furthermore, in comparison with other architectural and optimization interventions, we highlight DA's pivotal role as a data-centric method in addressing VRL's plasticity loss.

**A Factorial Examination of DA and Reset.** DA has become an indispensable component in achieving sample-efficient VRL applications (Yarats et al., 2020; Laskin et al., 2020b; Yarats et al., 2021a; Ma et al., 2022). As illustrated by the blue and orange dashed lines in Figure 1, employing a simple DA approach to the input observations can lead to significant performance improvements in previously unsuccessful algorithms. However, the mechanisms driving DA's notable effectiveness

remain largely unclear (Ma et al., 2024). On the other hand, recent studies have increasingly recognized that plasticity loss during training significantly hampers sample efficiency (Nikishin et al., 2022; Sokar et al., 2023; Lyle et al., 2023). This naturally raises the question: *does the remarkable efficacy of DA stem from its capacity to maintain plasticity?* To address this query, we undertake a factorial examination of DA and Reset. Given that Reset is well-recognized for its capability to mitigate the detrimental effects of plasticity loss on training, it can not only act as a diagnostic tool to assess the extent of plasticity loss in the presence or absence of DA, but also provide a benchmark to determine the DA's effectiveness in preserving plasticity.

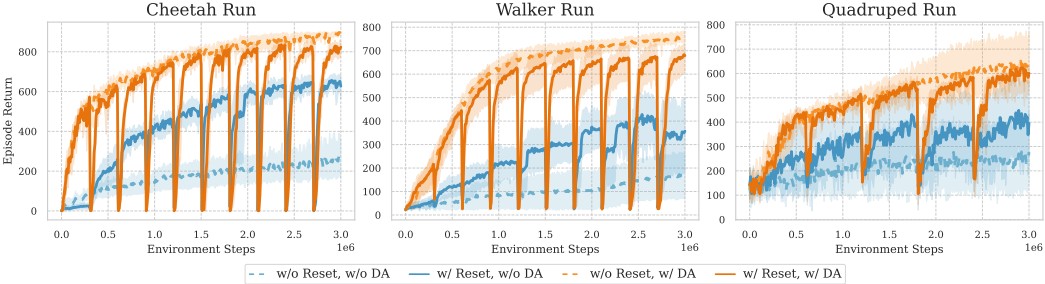

Figure 1: Training curves across four combinations: incorporating or excluding Reset and DA. We adopt DrQ-v2 (Yarats et al., 2021a) as our baseline algorithm and follow the Reset settings from Nikishin et al. (2022). Mean and std are estimated over 5 runs. Note that re-initializing 10 times in the Quadruped Run task resulted in poor performance, prompting us to adjust the reset times to 5. For ablation studies on reset times and results in other tasks, please refer to Appendix B.1.

The results presented in Figure 1 highlight three distinct phenomena: • In the absence of DA, the implementation of Reset consistently yields marked enhancements. This underscores the evident plasticity loss when training is conducted devoid of DA. • With the integration of DA, the introduction of Reset leads to only slight improvements, or occasionally, a decrease. This indicates that applying DA alone can sufficiently preserve the agent's plasticity, leaving little to no room for significant improvement. • Comparatively, the performance of Reset without DA lags behind that achieved employing DA alone, underscoring the potent effectiveness of DA in preserving plasticity.

**Comparing DA with Other Interventions.** We assess the influence of various architectural and optimization interventions on DMC using the DrQ-v2 framework. Specifically, we implement the following techniques: • *Weight Decay*, where we set the L2 coefficient to $10^{-5}$. • *L2 Init* (Kumar et al., 2023): This technique integrates L2 regularization aimed at the initial parameters into the loss function. Specifically, we apply it to the critic loss with a coefficient set to $10^{-2}$. • *Layer Normalization*(Ba et al., 2016) after each convolutional and linear layer. • *Spectral Normalization*(Miyato et al., 2018) after the initial linear layer for both the actor and

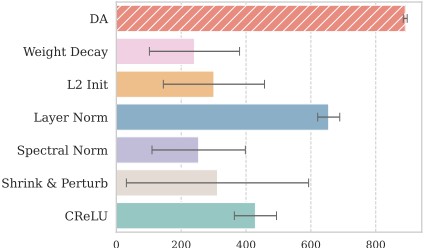

Figure 2: Performance of various interventions in Cheetah Run across 5 seeds.

critic networks. • *Shrink and Perturb*(Ash & Adams, 2020): This involves multiplying the critic network weights by a small scalar and adding a perturbation equivalent to the weights of a randomly initialized network. • Adoption of *CReLU* (Shang et al., 2016) as an alternative to ReLU in the critic network. We present the final performance of different interventions in Figure 2, which indicates that DA is the most effective method. For further comparison of interventions, see Appendix B.3.

## 4  MODULES: THE PLASTICITY LOSS OF CRITIC NETWORK IS PREDOMINANT

In this section, we aim to investigate *which module(s) of VRL agents suffer the most severe plasticity loss, and thus, are detrimental to efficient training*. Initially, by observing the differential trends in plasticity levels across modules with and without DA, we preliminarily pinpoint the critic's plasticity loss as the pivotal factor influencing training. Subsequently, the decisive role of DA when using a frozen pre-trained encoder attests that encoder's plasticity loss isn't the primary bottleneck for sample inefficiency. Finally, contrastive experiments with plasticity injection on actor and critic further corroborate that the critic's plasticity loss is the main culprit.

**Fraction of Active Units (FAU).** Although the complete mechanisms underlying plasticity loss remain unclear, a reduction in the number of active units within the network has been identified as a principal factor contributing to this deterioration (Lyle et al., 2023; Sokar et al., 2023; Lee et al., 2024). Hence, the Fraction of Active Units (FAU) is widely used as a metric for measuring plasticity. Specifically, the FAU for neurons located in module $\mathcal{M}$, denoted as $\Phi_{\mathcal{M}}$, is formally defined as:

$$\Phi_{\mathcal{M}} = \frac{\sum_{n \in \mathcal{M}} \mathbf{1}(a_n(x) > 0)}{N}, \tag{1}$$

where $a_n(x)$ represent the activation of neuron $n$ given the input $x$, and $N$ is the total number of neurons within module $\mathcal{M}$. More discussion on plasticity measurement can be found in Appendix A.

**Different FAU trends across modules reveal critic's plasticity loss as a hurdle for VRL training.** Within FAU as metric, we proceed to assess the plasticity disparities in the encoder, actor, and critic modules with and without DA. We adopt the experimental setup from Yarats et al. (2021a), where the encoder is updated only based on the critic loss. As shown in Figure 3 (left), the integration of DA leads to a substantial leap in training performance. Consistent with this uptrend in performance, DA elevates the critic's FAU to a level almost equivalent to an initialized network. In contrast, both the encoder and actor's FAU exhibit similar trends regardless of DA's presence or absence. This finding tentatively suggests that critic's plasticity loss is the bottleneck constraining training efficiency.

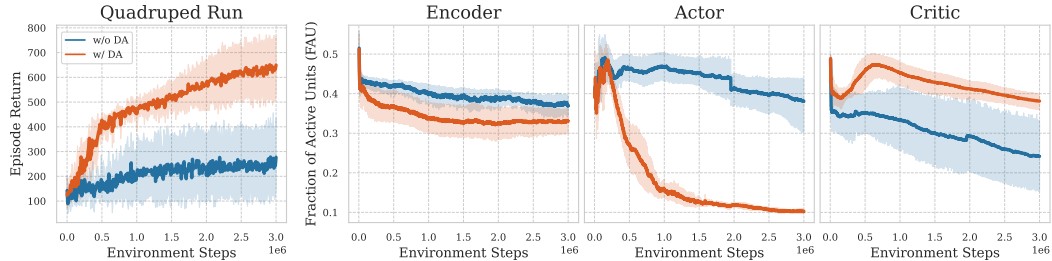

Figure 3: Different FAU trends across modules throughout training. The plasticity of encoder and actor displays similar trends whether DA is employed or not. Conversely, integrating DA leads to a marked improvement in the critic's plasticity. Further comparative results are in Appendix B.6.

**Is the sample inefficiency in VRL truly blamed on poor representation?** Since VRL handles high-dimensional image observations rather than well-structured states, prior studies commonly attribute VRL's sample inefficiency to its inherent challenge of learning a compact representation.

We contest this assumption by conducting a simple experiment. Instead of training the encoder from scratch, we employ an ImageNet pre-trained ResNet model as the agent's encoder and retain its parameters frozen throughout the training process. The specific implementation adheres Yuan et al. (2022), but employs the DA operation as in DrQ-v2. Building on this setup, we compare the effects of employing DA against not using it on sample efficiency, thereby isolating and negating the potential influences from disparities in the encoder's representation capability on training. As depicted in Figure 4, the results illustrate that employing DA consistently surpasses scenarios without DA by a notable margin.

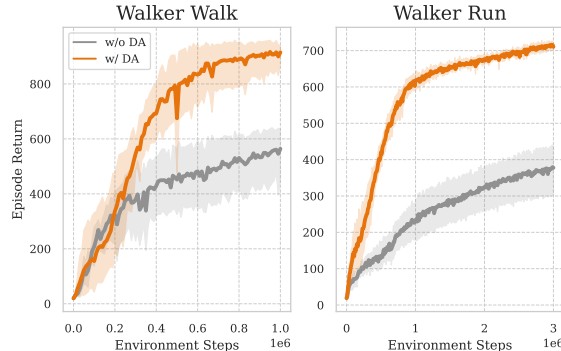

Figure 4: Learning curves of DrQ-v2 using a frozen ImageNet pre-trained encoder, with and without DA.

This significant gap sample inefficiency in VRL cannot be predominantly attributed to poor representation. This pronounced disparity underscores two critical insights: first, the pivotal effectiveness of DA is not centered on enhancing representation; and second, sample inefficiency in VRL cannot be primarily ascribed to the insufficient representation.

**Plasticity Injection on Actor and Critic as a Diagnostic Tool.** Having ruled out the encoder's influence, we next explore how the plasticity loss of actor and critic impact VRL's training efficiency.

To achieve this, we introduce plasticity injection as a diagnostic tool (Nikishin et al., 2024). Unlike Reset, which leads to a periodic momentary decrease in performance and induces an exploration effect (Nikishin et al., 2022), plasticity injection restores the module's plasticity to its initial level without altering other characteristics or compromising previously learned knowledge. Therefore, plasticity injection allows us to investigate in isolation the impact on training after reintroducing sufficient plasticity to a certain module. Should the training performance exhibit a marked enhancement relative to the baseline following plasticity injection into a particular module, it would suggest that this module had previously undergone catastrophic plasticity loss, thereby compromising the training efficacy. We apply plasticity injection separately to the actor and critic when us-

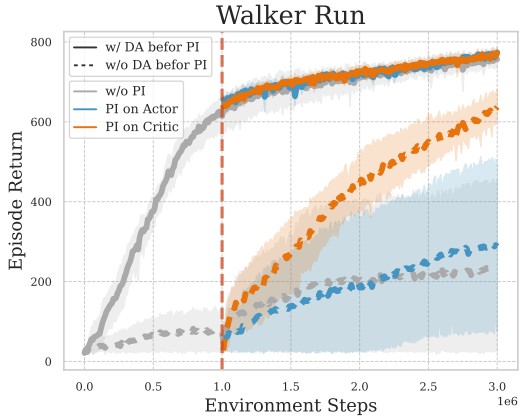

Figure 5: Training curves of employing plasticity injection (PI) on actor or critic. For all cases, DA is applied after the given environment steps.

ing and not using DA. The results illustrated in Figure 5 and Appendix B.4 reveal the subsequent findings and insights: ● When employing DA, the application of plasticity injection to both the actor and critic does not modify the training performance. This suggests that DA alone is sufficient to maintain plasticity within the Walker Run task. ● Without using DA in the initial 1M steps, administering plasticity injection to the critic resulted in a significant performance improvement. This fully demonstrates that the critic's plasticity loss is the primary culprit behind VRL's sample inefficiency.

## 5 STAGES: EARLY-STAGE PLASTICITY LOSS BECOMES IRRECOVERABLE

In this section, we elucidate the differential attributes of plasticity loss throughout various training stages. Upon confirming that the critic's plasticity loss is central to hampering training efficiency, a closer review of the results in Figure 3 underscores that DA's predominant contribution is to effectively recover the critic's plasticity during initial phases. This naturally raises two questions: ● After recovering the critic's plasticity to an adequate level in the early stage, will ceasing interventions to maintain plasticity detrimentally affect training? ● If interventions aren't applied early to recover the critic's plasticity, is it still feasible to enhance training performance later through such measures? To address these two questions, we conduct a comparative experiment by turning on or turning off DA at certain training steps and obtain the following findings: ● Turning off DA after the critic's plasticity has been recovered does not affect training efficiency. This suggests that it is not necessary to employ specific interventions to maintain plasticity in the later stages of training. ● Turning on DA when plasticity has already been significantly lost and without timely intervention in the early stages cannot revive the agent's training performance. This observation underscores the vital importance of maintaining plasticity in the early stages; otherwise, the loss becomes irrecoverable.

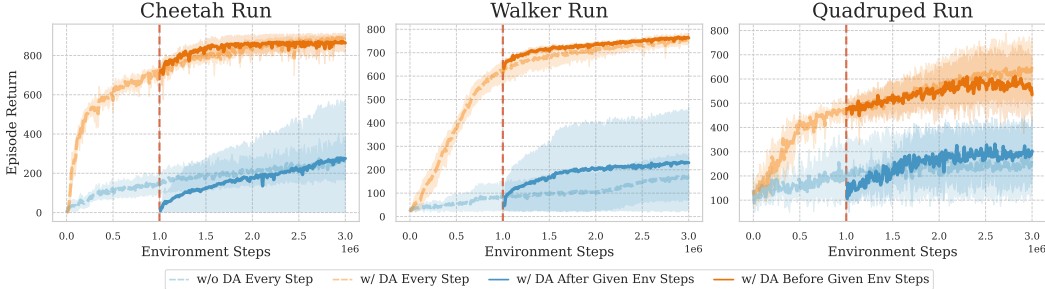

Figure 6: Training curves for various DA application modes. The red dashed line shows when DA is turned on or turned off. Additional comparative results can be found in Appendix B.5.

We attribute these differences across stages to the nature of online RL to learn from scratch in a bootstrapped fashion. During the initial phases of training, bootstrapped target derived from low-quality and limited-quantity experiences exhibits high non-stationarity and deviates significantly from the

actual state-action values (Cetin et al., 2022). The severe non-stationarity of targets induces a rapid decline in the critic's plasticity (Lee et al., 2024; Li et al., 2022a), consistent with the findings in Figure 3. Having lost the ability to learn from newly collected data, the critic will perpetually fail to capture the dynamics of the environment, preventing the agent from acquiring an effective policy. This leads to ***catastrophic plasticity loss*** in the early stages. Conversely, although the critic's plasticity experiences a gradual decline after recovery, this can be viewed as a process of progressively approximating the optimal value function for the current task. For single-task VRL that doesn't require the agent to retain continuous learning capabilities, this is termed as a ***benign plasticity loss***. Differences across stages offer a new perspective to address VRL's plasticity loss challenges.

## 6    METHODS: ADAPTIVE RR FOR ADDRESSING THE HIGH RR DILEMMA

Drawing upon the refined understanding of plasticity loss, this section introduces *Adaptive RR* to tackle the high RR dilemma in VRL. Extensive evaluations demonstrate that *Adaptive RR* strikes a superior trade-off between reuse frequency and plasticity loss, thereby improving sample efficiency.

**High RR Dilemma.** Increasing the replay ratio (RR), which denotes the number of updates per environment interaction, is an intuitive strategy to further harness the strengths of off-policy algorithms to improve sample efficiency. However, recent studies (D'Oro et al., 2022; Li et al., 2022a) and the results in Figure 7 consistently reveal that adopting a higher static RR impairs training efficiency.

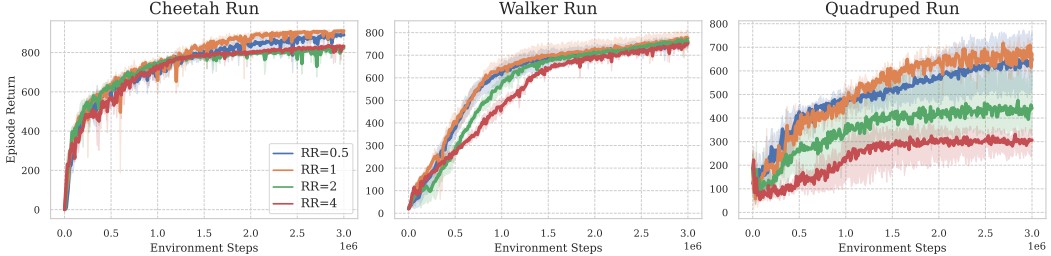

Figure 7: Training curves across varying RR values. Despite its intent to enhance sample efficiency through more frequent updates, an increasing RR value actually undermines training.

The fundamental mechanism behind this counterintuitive failure of high RR is widely recognized as the intensified plasticity loss (Nikishin et al., 2022; Sokar et al., 2023). As illustrated in Figure 8, increasing RR results in a progressively exacerbated plasticity loss during the early stages of training. Increasing RR from $0.5$ to $1$ notably diminishes early-stage plasticity, but the heightened reuse frequency compensates, resulting in a marginal boost in sample efficiency. However, as RR continues to rise, the detrimental effects of plasticity loss become predominant, leading to a consistent decline in sample efficiency. When RR increases to $4$, even with the intervention of DA, there's no discernible recovery of the critic's plasticity in the early stages, culminating in a catastrophic loss.

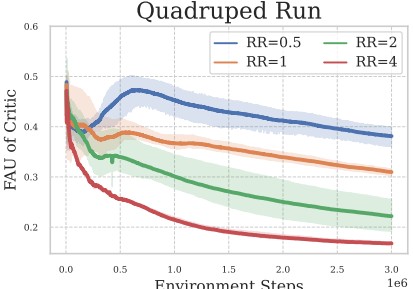

Figure 8: The FAU of critic across varying RR values. A larger RR value leads to more severe plasticity loss.

An evident high RR dilemma quandary arises: while higher reuse frequency holds potential for improving sample efficiency, the exacerbated plasticity loss hinders this improvement.

**Can we adapt RR instead of setting a static value?** Previous studies addressing the high RR dilemma typically implement interventions to mitigate plasticity loss while maintaining a consistently high RR value throughout training (D'Oro et al., 2022; Sokar et al., 2023; Nikishin et al., 2024). Drawing inspiration from the insights in Section 5, which highlight the varying plasticity loss characteristics across different training stages, an orthogonal approach emerges: ***why not dynamically adjust the RR value based on the current stage?*** Initially, a low RR is adopted to prevent catastrophic plasticity loss. In later training stages, RR can be raised to boost reuse frequency, as the plasticity dynamics become benign. This balance allows us to sidestep early high RR drawbacks and later harness the enhanced sample efficiency from greater reuse frequency. Furthermore, the observations in Section 4 have confirmed that the critic's plasticity, as measured by its FAU, is the primary factor influencing sample efficiency. This implies that the FAU of critic module can

be employed adaptively to identify the current training stage. Once the critic's FAU has recovered to a satisfactory level, it indicates the agent has moved beyond the early training phase prone to catastrophic plasticity loss, allowing for an increase in the RR value. Based on these findings and considerations, we propose our method, *Adaptive RR*.

---
**Adaptive Replay Ratio**

*Adaptive RR* adjusts the ratio according to the current plasticity level of critic, utilizing a low RR in the early stage and transitioning it to a high value after the plasticity recovery stages.

---

**Evaluation on DeepMind Control Suite.** We then evaluate the effectiveness of *Adaptive RR* ↻ in improving the sample efficiency of VRL algorithms. Our experiments are conducted on six challenging continuous DMC tasks, which are widely perceived to significantly suffer from plasticity loss (Lee et al., 2024). We select two static RR values as baselines: ● Low RR=0.5, which exhibits no severe plasticity loss but has room for improvement in its reuse frequency. ● High RR=2, which shows evident plasticity loss, thereby damaging sample efficiency. Our method, *Adaptive RR*, starts with RR=0.5 during the initial phase of training, aligning with the default setting of DrQ-v2. Subsequently, we monitor the FAU of the critic module every 50 episodes, *i.e.*, $5 \times 10^4$ environment steps. When the FAU difference between consecutive checkpoints drops below a minimal threshold (set at $0.001$ in our experiments), marking the end of the early stage, we adjust the RR to $2$. Figure 9 illustrates the comparative performances. *Adaptive RR* consistently demonstrates superior sample efficiency compared to a static RR throughout training.

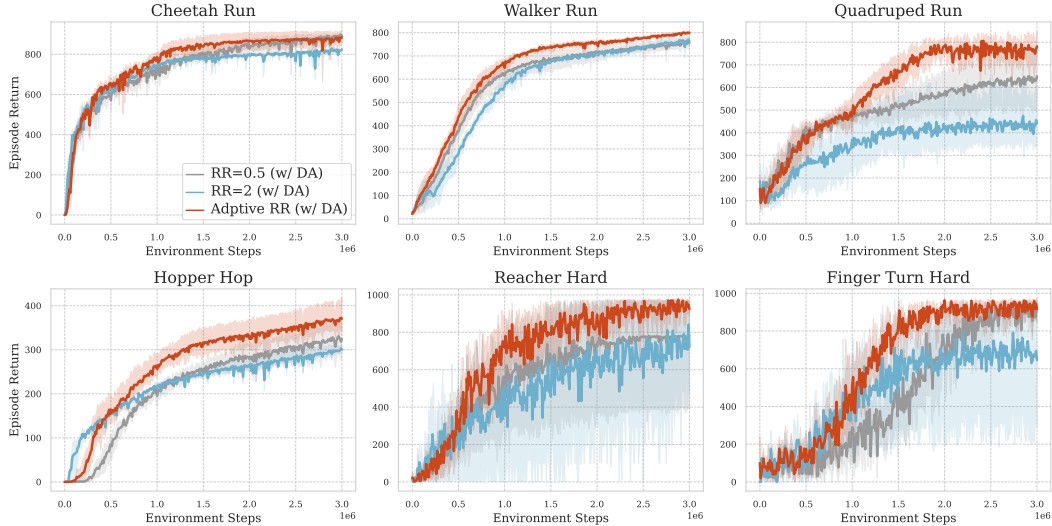

Figure 9: Training curves of various RR settings across 6 challenging DMC tasks. *Adaptive RR* demonstrates superior sample efficiency compared to both static low RR and high RR value.

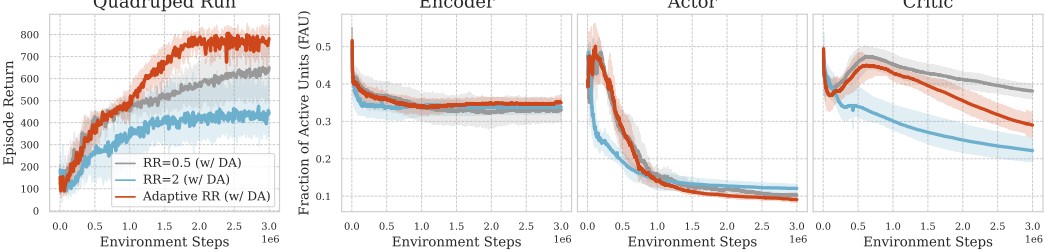

Figure 10: Evolution of FAU across the three modules in the Quadruped Run task under different RR configurations. The critic's plasticity is most influenced by different RR settings. Under high RR, the critic's plasticity struggles to recover in early training. In contrast, *Adaptive RR* successfully mitigates catastrophic plasticity loss in the early phases, yielding the optimal sample efficiency.

Through a case study on Quadruped Run, we delve into the underlying mechanism of *Adaptive RR*, as illustrated in Figure 10. Due to the initial RR set at $0.5$, the critic's plasticity recovers promptly to

---
↻Our code is available at: https://github.com/Guozheng-Ma/Adaptive-Replay-Ratio

a considerable level in the early stages of training, preventing catastrophic plasticity loss as seen with RR=2. After the recovery phases, *Adaptive RR* detects a slow change in the critic's FAU, indicating it's nearing a peak, then switch the RR to a higher value. Within our experiments, the switch times for the five seeds occurred at: 0.9M, 1.2M, 1.1M, 0.95M, and 0.55M. After switching to RR=2, the increased update frequency results in higher sample efficiency and faster convergence rates. Even though the critic's FAU experiences a more rapid decline at this stage, this benign loss doesn't damage training. Hence, *Adaptive RR* can effectively exploits the sample efficiency advantages of a high RR, while adeptly avoiding the severe consequences of increased plasticity loss.

Table 1: Comparison of Adaptive RR versus static RR with Reset and ReDo implementations. The average episode returns are averaged over 5 seeds after training for 2M environment steps.

| Average Episode Return | RR=0.5 | | | RR=2 | | | Adaptive RR |
|---|---|---|---|---|---|---|---|
| (After 2M Env Steps) | default | Reset | ReDo | default | Reset | ReDo | RR:0.5to2 |
| Cheetah Run | $828 \pm 59$ | $799 \pm 26$ | $788 \pm 5$ | $793 \pm 9$ | $\mathbf{885 \pm 20}$ | $873 \pm 19$ | $880 \pm 45$ |
| Walker Run | $710 \pm 39$ | $648 \pm 107$ | $618 \pm 50$ | $709 \pm 7$ | $749 \pm 10$ | $734 \pm 16$ | $\mathbf{758 \pm 12}$ |
| Quadruped Run | $579 \pm 120$ | $593 \pm 129$ | $371 \pm 158$ | $417 \pm 110$ | $511 \pm 47$ | $608 \pm 53$ | $\mathbf{784 \pm 53}$ |

We further compare the performance of Adaptive RR with that of employing Reset (Nikishin et al., 2022) and ReDo (Sokar et al., 2023) under static RR conditions. Although Reset and ReDo both effectively mitigate plasticity loss in high RR scenarios, our method significantly outperforms these two approaches, as demonstrated in Table 1 This not only showcases that Adaptive RR can secure a superior balance between reuse frequency and plasticity loss but also illuminates the promise of dynamically modulating RR in accordance with the critic's overall plasticity level as an effective strategy, alongside neuron-level network parameter resetting, to mitigate plasticity loss.

**Evaluation on Atari-100k.** To demonstrate the applicability of Adaptive RR in discrete-action tasks we move our evaluation to the Atari-100K benchmark (Kaiser et al., 2019), assessing Adaptive RR against three distinct static RR strategies across 17 games. In static RR settings, as shown in Table 2, algorithm performance significantly declines when RR increases to 2, indicating that the negative impact of plasticity loss gradually become dominant.

Table 2: Summary of Atari-100K results. Comprehensive scores are available in Appendix D.

| *Metrics* | DrQ($\epsilon$) | | | ReDo | Adaptive RR |
|---|---|---|---|---|---|
| | RR=0.5 | RR=1 | RR=2 | RR=1 | RR:0.5to2 |
| *Mean HNS* (%) | 42.3 | 41.3 | 35.1 | 42.3 | **55.8** |
| *Median HNS* (%) | 22.6 | 30.3 | 26.0 | 41.6 | **48.7** |
| *# Superhuman* | 3 | 1 | 1 | 2 | **4** |
| *# Best* | 0 | 2 | 1 | 3 | **11** |

However, Adaptive RR, by appropriately increasing RR from 0.5 to 2 at the right moment, can effectively avoid catastrophic plasticity loss, thus outperforming other configurations in most tasks.

## 7 CONCLUSION, LIMITATIONS, AND FUTURE WORK

In this work, we delve deeper into the plasticity of VRL, focusing on three previously underexplored aspects, deriving pivotal and enlightening insights: • DA emerges as a potent strategy to mitigate plasticity loss. • Critic's plasticity loss stands as the primary hurdle to the sample-efficient VRL. • Ensuring plasticity recovery during the early stages is pivotal for efficient training. Armed with these insights, we propose *Adaptive RR* to address the high RR dilemma that has perplexed the VRL community for a long time. By striking a judicious balance between sample reuse frequency and plasticity loss management, *Adaptive RR* markedly improves the VRL's sample efficiency.

**Limitations.** Firstly, our experiments focus on DMC and Atari environments, without evaluation in more complex settings. As task complexity escalates, the significance and difficulty of maintaining plasticity concurrently correspondingly rise. Secondly, we only demonstrate the effectiveness of *Adaptive RR* under basic configurations. A more nuanced design could further unlock its potential.

**Future Work.** Although neural networks have enabled scaling RL to complex decision-making scenarios, they also introduce numerous difficulties unique to DRL, which are absent in traditional RL contexts. Plasticity loss stands as a prime example of these challenges, fundamentally stemming from the contradiction between the trial-and-error nature of RL and the inability of neural networks to continuously learn non-stationary targets. To advance the real-world deployment of DRL, it is imperative to address and understand its distinct challenges. Given RL's unique learning dynamics, exploration of DRL-specific network architectures and optimization techniques is essential.

ACKNOWLEDGEMENTS

This work is supported by STI 2030—Major Projects (No. 2021ZD0201405). We thank Zilin Wang and Haoyu Wang for their valuable suggestions and collaboration. We also extend our thanks to Evgenii Nikishin for his support in the implementation of plasticity injection. Furthermore, we sincerely appreciate the time and effort invested by the anonymous reviewers in evaluating our work, and are grateful for their valuable and insightful feedback.

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

## A    EXTENDED RELATED WORK

In this section, we provide an extended related work to supplement the related work presented in the main body.

**Sample-Efficient VRL.**    Prohibitive sample complexity has been identified as the primary obstacle hindering the real-world applications of VRL (Yarats et al., 2021b). Previous studies ascribe this inefficiency to VRL's requirements to concurrently optimize task-specific policies and learn compact state representations from high-dimensional observations. As a result, significant efforts have been directed towards improving sample efficiency through the training of a more potent *encoder*. The most representative approaches design *auxiliary representation tasks* to complement the RL objective, including pixel or latent reconstruction (Yarats et al., 2021b; Yu et al., 2022), future prediction (Lee et al., 2020; Schwarzer et al., 2020; Yu et al., 2021), and contrastive learning for instance (Laskin et al., 2020a; Sun et al., 2022; Fan & Li, 2022) or temporal discrimination (Zhu et al., 2022; Oord et al., 2018; Nguyen et al., 2021; Mazoure et al., 2020). Another approach is to *pre-train a visual encoder* that enables efficient adaptation to downstream tasks (Shah & Kumar, 2021; Xiao et al., 2022; Parisi et al., 2022; Nair et al., 2023). However, recent empirical studies suggest that these methods do not consistently improve training efficiency (Li et al., 2022b; Hansen et al., 2023; Ma et al., 2024), indicating that insufficient representation may not be the primary bottleneck hindering the sample efficiency of current algorithms. Our findings in Section 4 provide a compelling explanation for the limited impact of enhanced representation: the plasticity loss within the critic module is the primary constraint on VRL's sample efficiency.

**Plasticity Loss in Continual Learning vs. in Reinforcement Learning.**    Continual Learning (CL) aims to continuously acquire new tasks, referred to as plasticity, without forgetting previously learned tasks, termed stability. A primary challenge in CL is managing the stability-plasticity trade-off. Although online reinforcement learning (RL) exhibits characteristics of plasticity due to its non-stationary learning targets, there are fundamental differences between CL and RL. Firstly, online RL typically begins its learning process from scratch, which can lead to limited training data in the early stages. This scarcity of data can subsequently result in a loss of plasticity early on. Secondly, RL usually doesn't require an agent to learn multiple policies. Therefore, any decline in plasticity during the later stages won't significantly impact its overall performance.

**Measurement Metrics of Plasticity.**    Several metrics are available to assess plasticity, including weight norm, feature rank, visualization of loss landscape, and the fraction of active units (FAU). The weight norm (commonly of both encoder and head) serves a dual purpose: it not only acts as a criterion to determine when to maintain plasticity but also offers a direct method to regulate plasticity through L2 regularization (Sokar et al., 2023; Nikishin et al., 2024). However, Nikishin et al. (2024) show that the weight norm is sensitive to environments and cannot address the plasticity by controlling itself. The feature rank can be also regarded as a proxy metric for plasticity loss (Kumar et al., 2020; Lyle et al., 2021). Although the feature matrices used by these two works are slightly different, they correlate the feature rank with performance collapse. Nevertheless, Gulcehre et al. (2022) observe that the correlation appears in restricted settings. Furthermore, the loss landscape has been drawing increasing attention for its ability to directly reflect the gradients in backpropagation. Still, computing the network's Hessian concerning a loss function and the gradient covariance can be computationally demanding (Lyle et al., 2023). Our proposed method aims to obtain a reliable criterion without too much additional computation cost, and leverage it to guide the plasticity maintenance. We thus settled on the widely-recognized and potent metric, FAU, for assessing plasticity (Sokar et al., 2023; Abbas et al., 2023). This metric provides an upper limit on the count of inactive units. As shown in Figure 9, the experimental results validate that A-RR based on FAU significantly outperforms static RR baselines. Although FAU's efficacy is evident in various studies, including ours, its limitations in convolutional networks are highlighted by (Lyle et al., 2023). Therefore, we advocate for future work to introduce a comprehensive and resilient plasticity metric.

# B EXTENDED EXPERIMENT RESULTS

## B.1 RESET

To enhance the plasticity of the agent's network, the *Reset* method periodically re-initializes the parameters of its last few layers, while preserving the replay buffer. In Figure 11, we present additional experiments on six DMC tasks, exploring four scenarios: with and without the inclusion of both Reset and DA. Although reset is widely acknowledged for its efficacy in counteracting the adverse impacts of plasticity loss, our findings suggest its effectiveness largely hinges on the hyper-parameters determining the reset interval, as depicted in Figure 12.

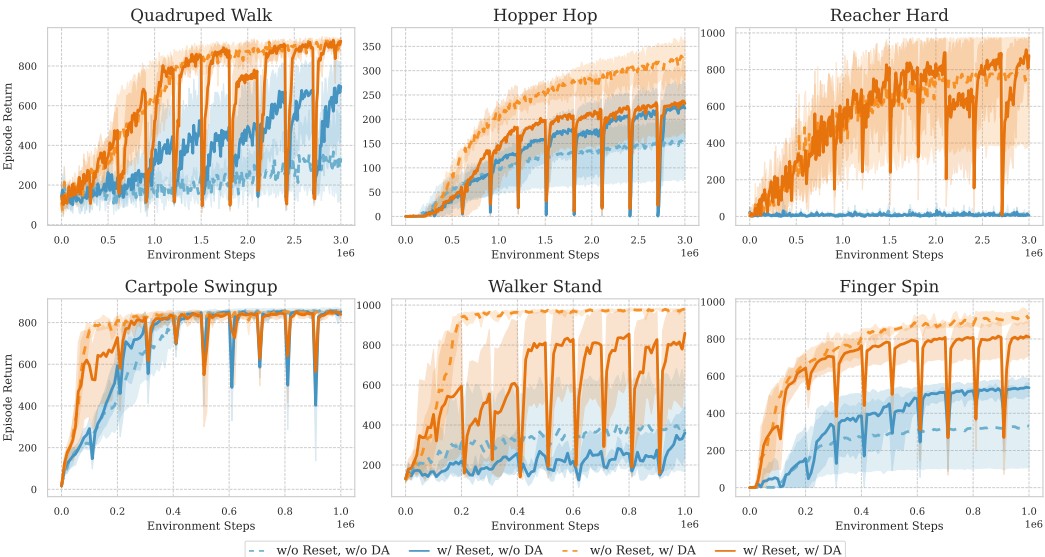

Figure 11: Training curves across four combinations: incorporating or excluding Reset and DA.

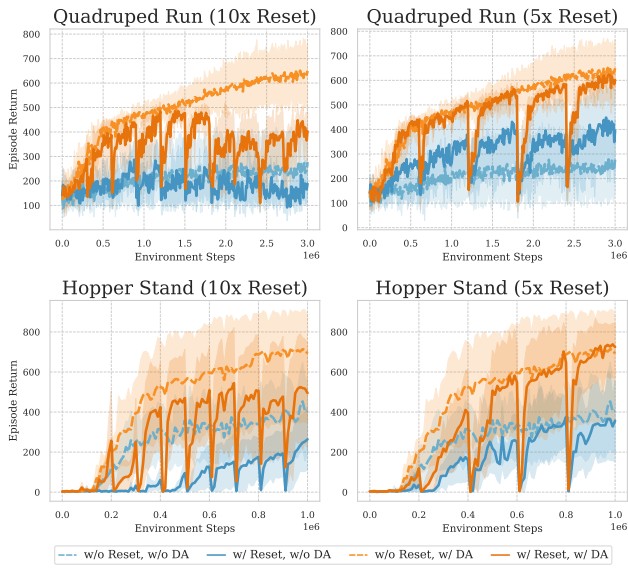

Figure 12: Learning curves for various reset intervals demonstrate that the effect of the reset strongly depends on the hyper-parameter that determines the reset interval.

## B.2 HEAVY PRIMING PHENOMENON

Heavy priming (Nikishin et al., 2022) refers to updating the agent $10^5$ times using the replay buffer, which collects 2000 transitions after the start of the training process. Heavy priming can induce the agent to overfit to its early experiences. We conducted experiments to assess the effects of using heavy priming and DA, both individually and in combination. The training curves can be found in Figure 13. The findings indicate that, while heavy priming can markedly impair sample efficiency without DA, its detrimental impact is largely mitigated when DA is employed. Additionally, we examine the effects of employing DA both during heavy priming and subsequent training, as illustrated in Figure 14. The results indicate that DA not only mitigates plasticity loss during heavy priming but also facilitates its recovery afterward.

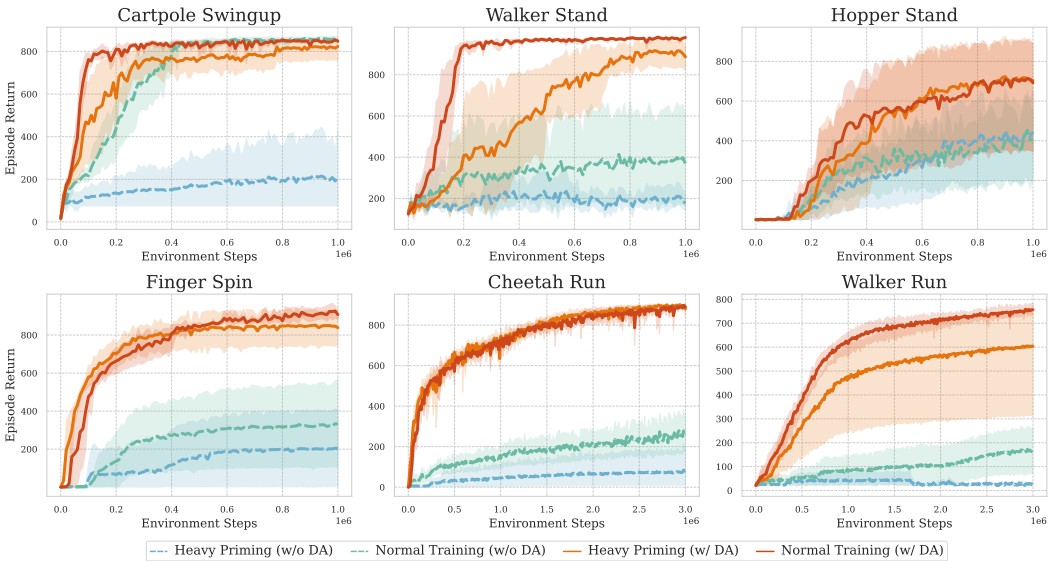

Figure 13: Heavy priming can severely damage the training efficiency when not employing DA.

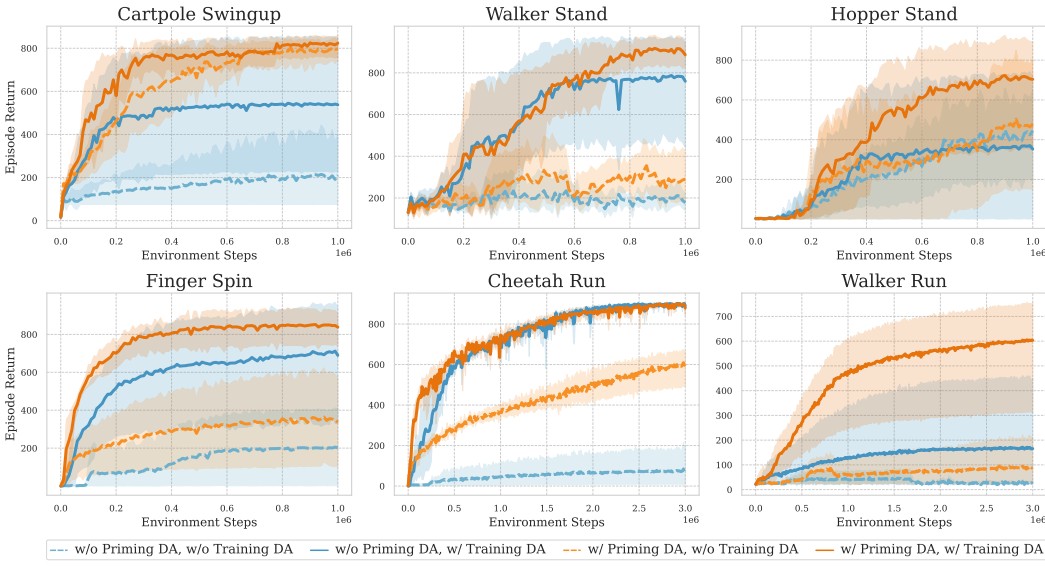

Figure 14: DA not only can prevent the plasticity loss but also can recover the plasticity of agent after heavy priming phase.

## B.3   FURTHER COMPARISONS OF DA WITH OTHER INTERVENTIONS

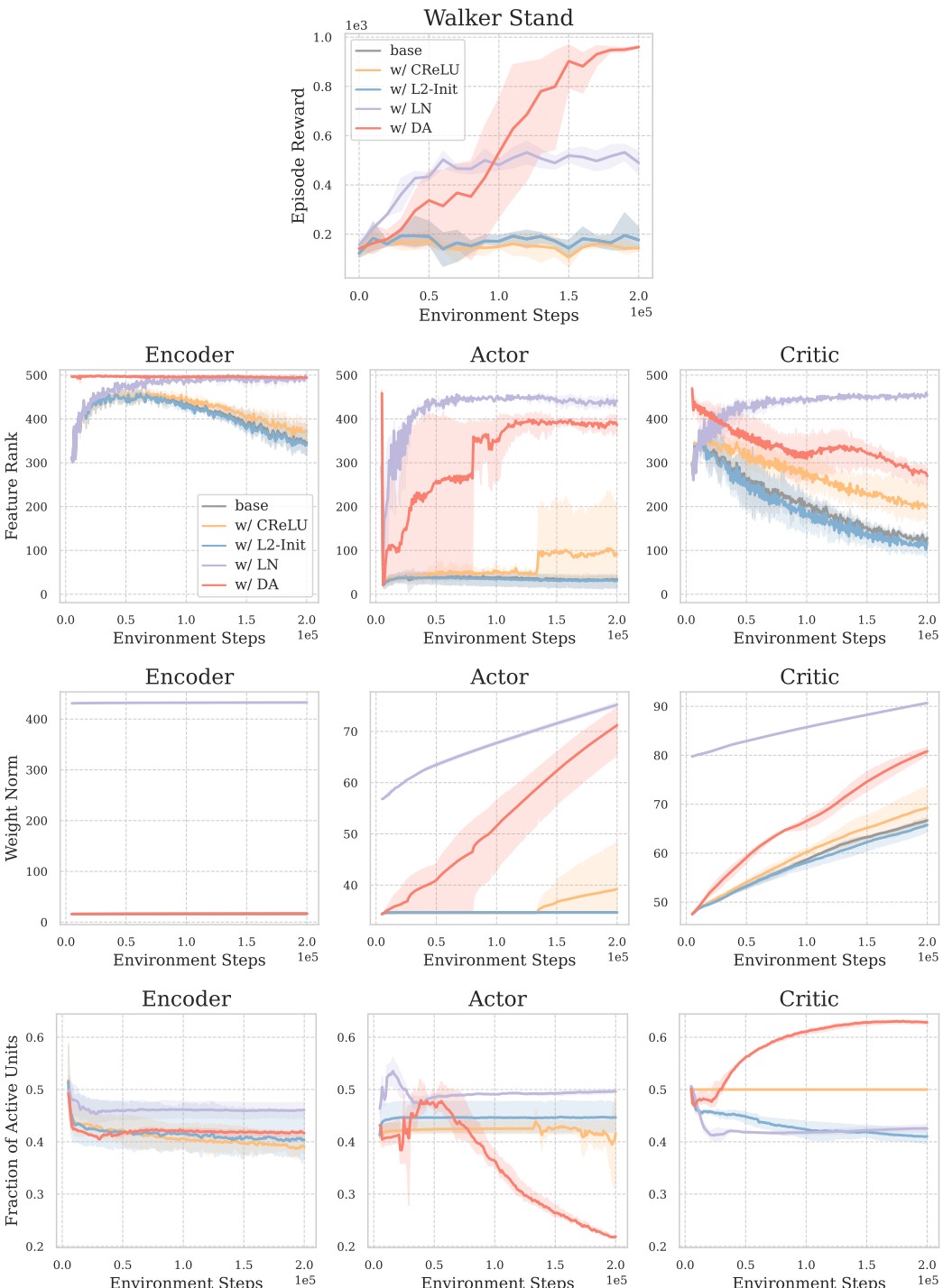

Figure 15: We use three metrics - Feature Rank, Weight Norm, and Fraction of Active Units (FAU) - to assess the impact of various interventions on training dynamics.

### B.4 PLASTICITY INJECTION

*Plasticity injection* is an intervention to increase the plasticity of a neural network. The network is schematically separated into an encoder $\phi(\cdot)$ and a head $h_\theta(\cdot)$. After plasticity injection, the parameters of head $\theta$ are frozen. Subsequently, two randomly initialized parameters, $\theta_1'$ and $\theta_2'$ are created. Here, $\theta_1'$ are trainable and $\theta_2'$ are frozen. The output from the head is computed using the formula $h_\theta(z) + h_{\theta_1'}(z) - h_{\theta_2'}(z)$, where $z = \phi(x)$.

We conducted additional plasticity injection experiments on Cheetah Run and Quadruped Run within DMC, as depicted in Figure 16. The results further bolster the conclusion made in Section 4: critic's plasticity loss is the primary culprit behind VRL's sample inefficiency.

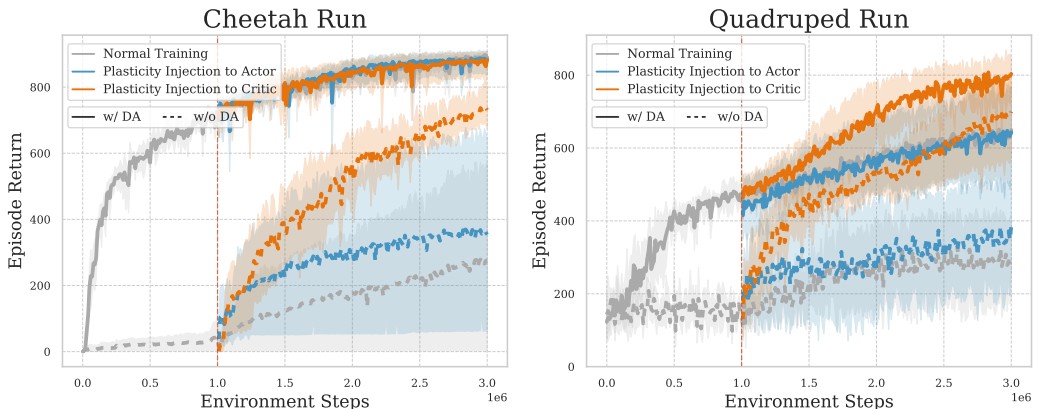

Figure 16: Training curves showcasing the effects of Plasticity Injection on either the actor or critic, evaluated on Cheetah Run and Quadruped Run.

### B.5 TURNING ON OR TURNING OFF DA AT EARLY STAGES

In Figure 17, we present additional results across six DMC tasks for various DA application modes. As emphasized in Section 5, it is necessary to maintain plasticity during early stages.

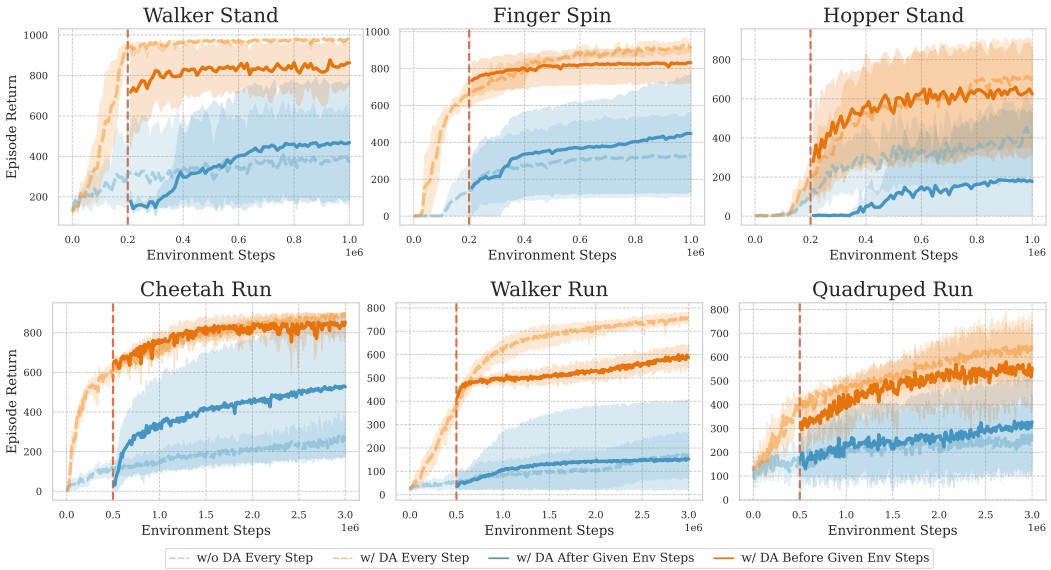

Figure 17: Training curves across different DA application modes, illustrating the critical role of plasticity in the early stage.

## B.6 FAU Trends Across Different Tasks

In Figure 18 and Figure 19, we showcase trends for various FAU modules across an additional nine DMC tasks as a complement to the main text.

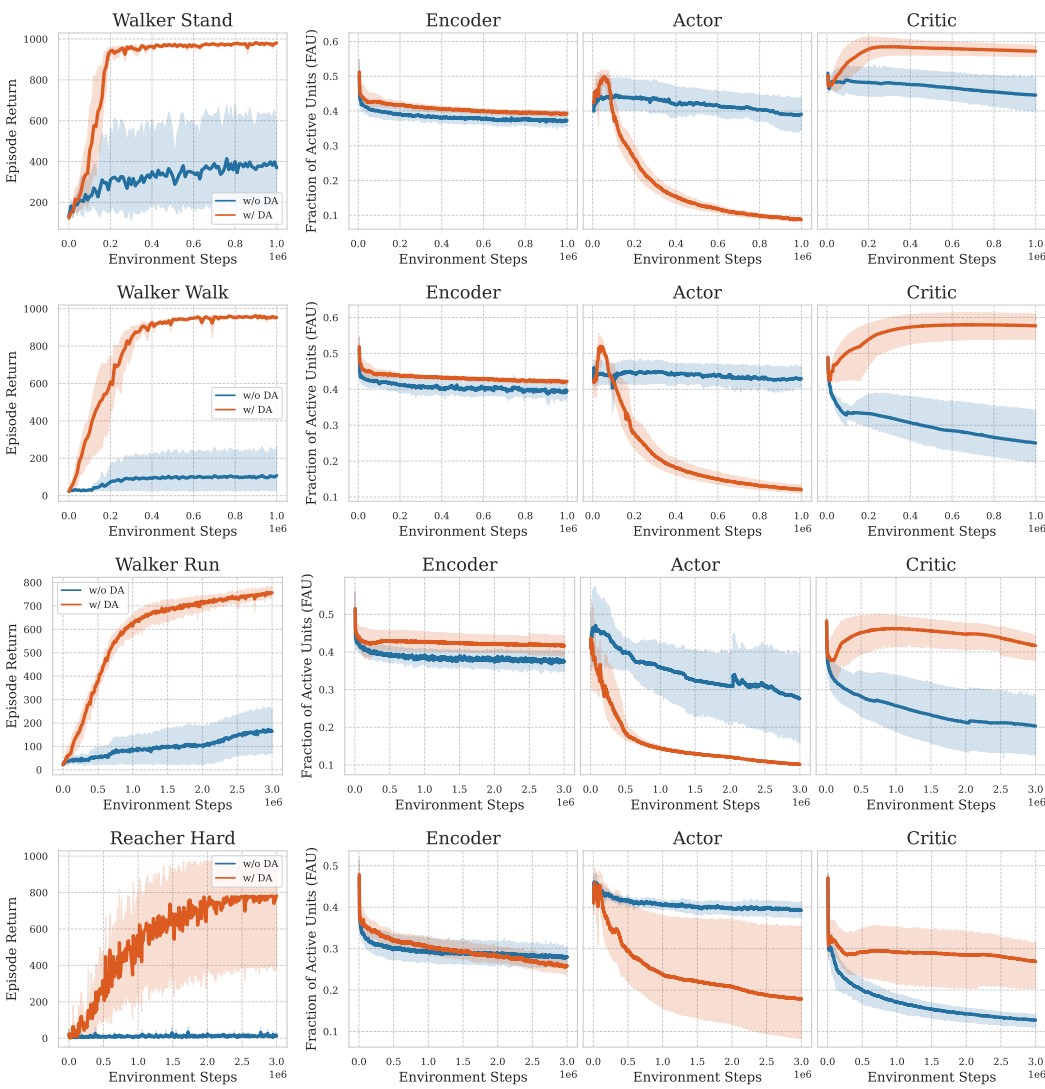

Figure 18: FAU trends for various modules within the VRL agent across DMC tasks (*Walker Stand*, *Walker Walk*, *Walker Run* and *Reacher Hard*) throughout the training process.

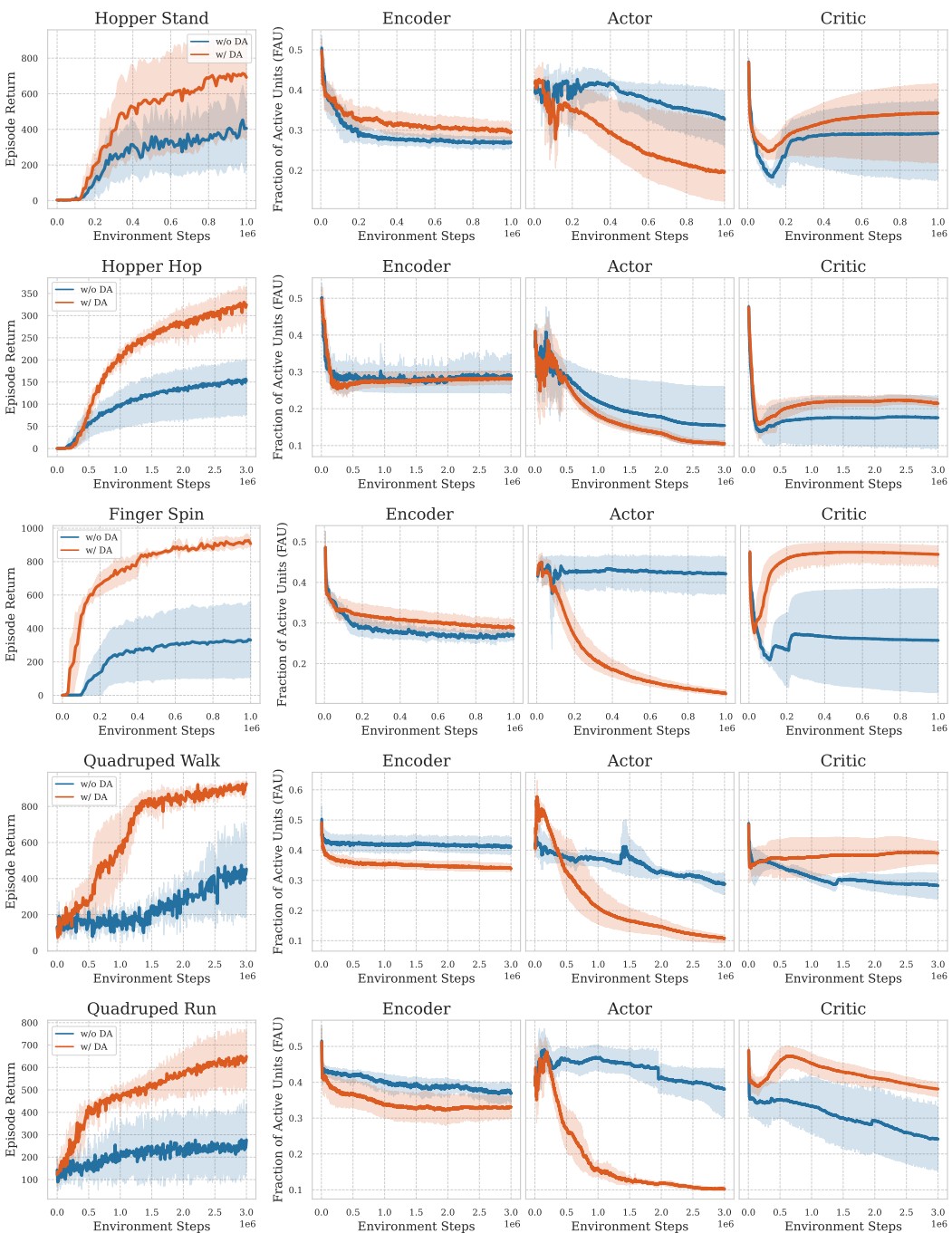

Figure 19: FAU trends for various modules within the VRL agent across DMC tasks (*Hopper Stand*, *Hopper Hop*, *Finger Spin*, *Quadruped Walk* and *Quadruped Run*) throughout the training process.

Figure 20 provides the FAU for different random seeds, demonstrating that the trend is consistent across all random seeds.

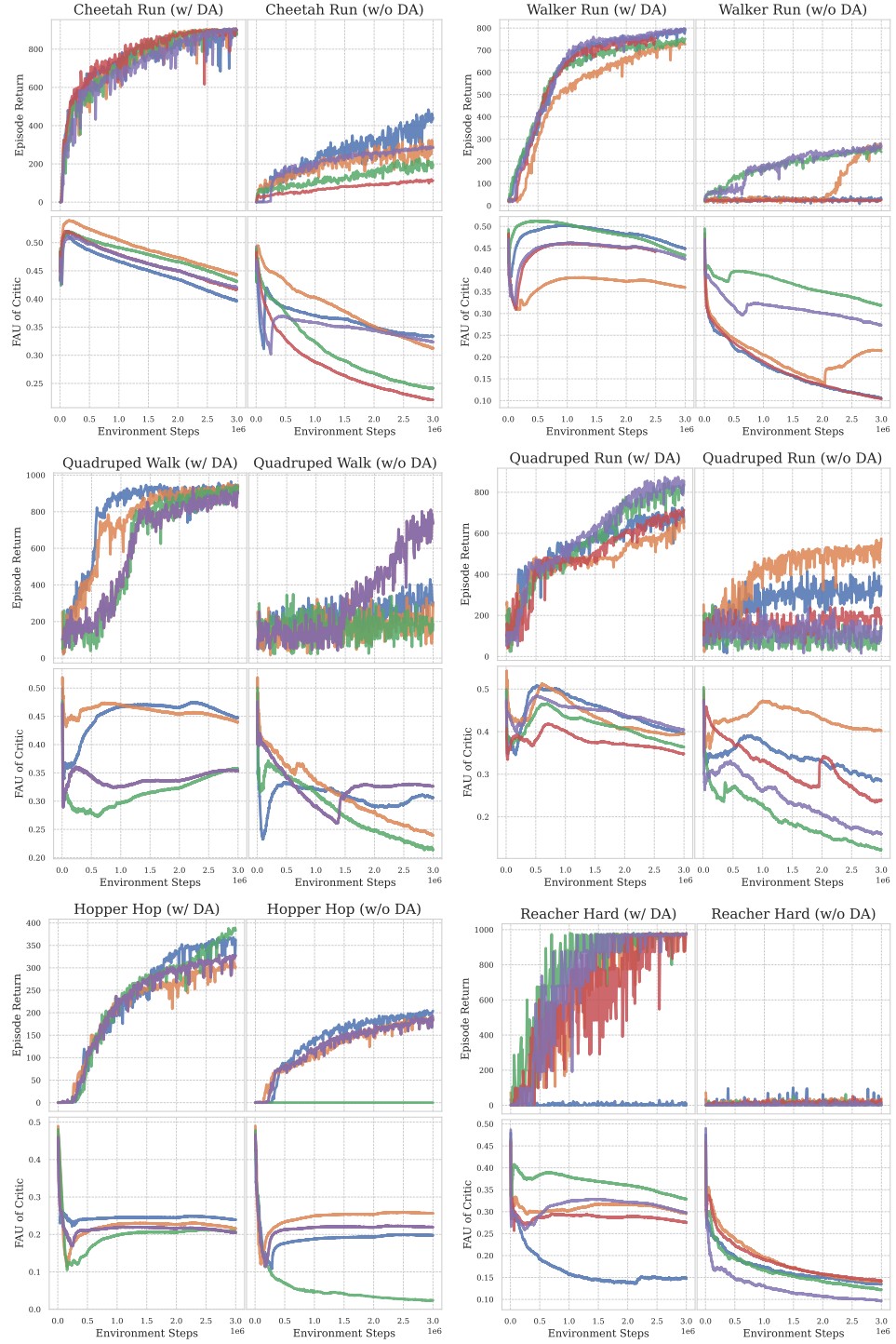

Figure 20: FAU trends for various modules within the VRL agent, evaluated across six DMC tasks and observed for each random seed throughout the training process.

## B.7   ADDITIONAL METRICS TO QUANTIFY THE PLASTICITY

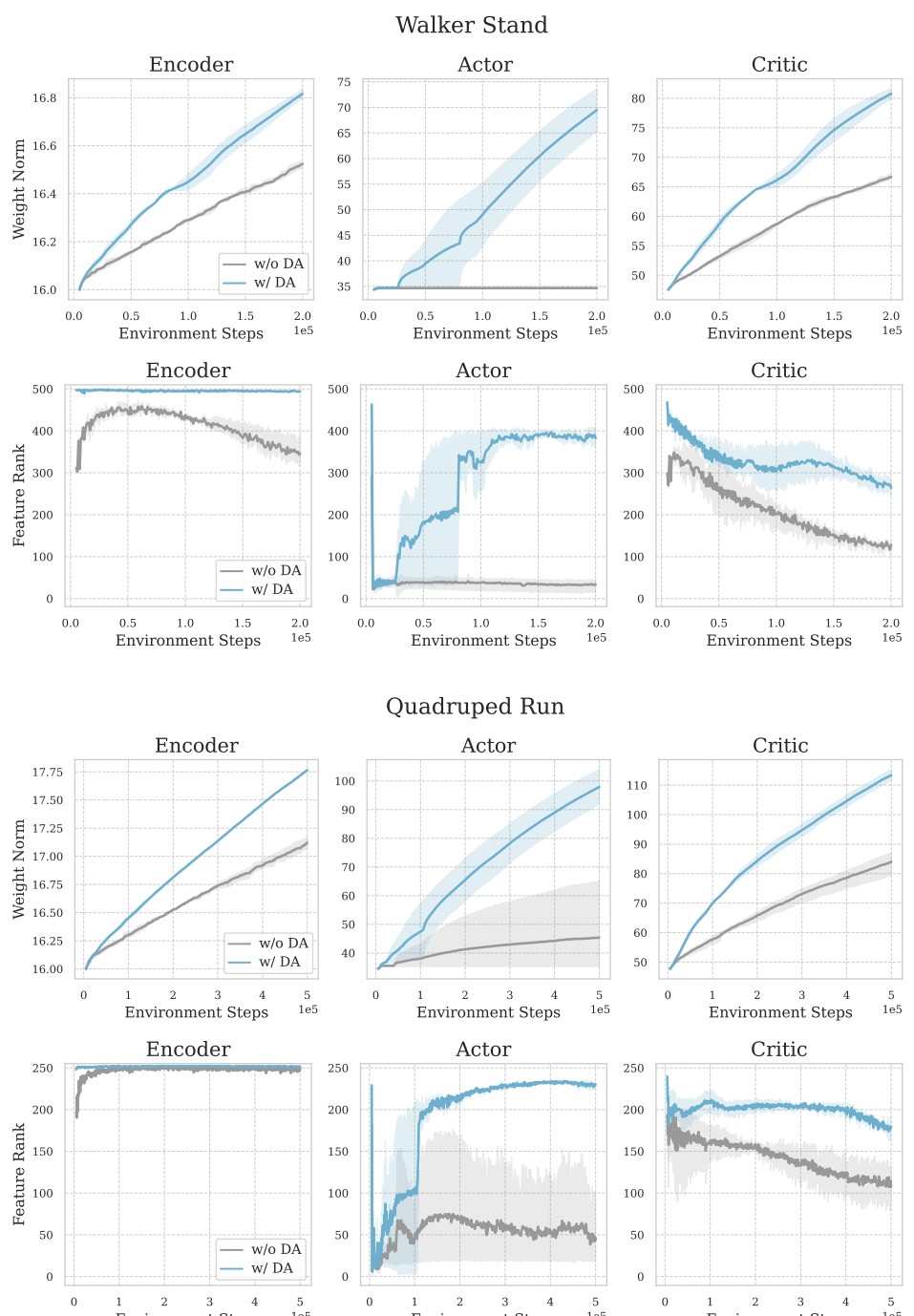

Figure 21: Measuring the plasticity of different modules via feature rank and weight norm.

## C EXPERIMENTAL DETAILS

In this section, we provide our detailed setting in experiments.

### C.1 ALGORITHM

---
**Algorithm 1** Adaptive RR
---
**Require:** Check interval $I$, threshold $\tau$, total steps $T$
  1: Initialize RL training with a low RR
  2: **while** $t < T$ **do**
  3:   **if** $t\%I = 0$ and $|\Phi_C^t - \Phi_C^{t-I}| < \tau$ **then**
  4:       Switch to high RR
  5:   **end if**
  6:   Continue RL training with the current RR
  7:   Increment step $t$
  8: **end while**
---

### C.2 DMC SETUP

We conducted experiments on robot control tasks within DeepMind Control using image input as the observation. All experiments are based on previously superior DrQ-v2 algorithms and maintain all hyper-parameters from DrQ-v2 unchanged. The only modification made was to the replay ratio, adjusted according to the specific setting. The hyper-parameters are presented in Table 3.

Table 3: A default set of hyper-parameters used in DMControl evaluation.

| Algorithms Hyper-parameters | |
|---|---|
| Replay buffer capacity | $10^6$ |
| Action repeat | 2 |
| Seed frames | 4000 |
| Exploration steps | 2000 |
| $n$-step returns | 3 |
| Mini-batch size | 256 |
| Discount $\gamma$ | 0.99 |
| Optimizer | Adam |
| Learning rate | $10^{-4}$ |
| Critic Q-function soft-update rate $\tau$ | 0.01 |
| Features dim. | 50 |
| Repr. dim. | $32 \times 35 \times 35$ |
| Hidden dim. | 1024 |
| Exploration stddev. clip | 0.3 |
| Exploration stddev. schedule | $\mathrm{linear}(1.0, 0.1, 500000)$ |

# D  EVALUATION ON ATARI

**Implement details.** Our Atari experiments and implementation were based on the Dopamine framework (Castro et al., 2018). For ReDo (Sokar et al., 2023) and DrQ($\epsilon$), We used the same setting as Dopamine, shown in Table 4. we use 5 independent random seeds for each Atari game. The detailed results are shown in Table 5.

Table 4: Hyper-parameters for Atari-100K.

| Common Parameter-DrQ($\epsilon$) | Value |
|---|---|
| Optimizer | Adam |
| Optimizer: Learning rate | $1 \times 10^{-4}$ |
| Optimizer: $\epsilon$ | $1.5 \times 10^{-4}$ |
| Training $\epsilon$ | 0.01 |
| Evaluation $\epsilon$ | 0.001 |
| Discount factor | 0.99 |
| Replay buffer size | $10^6$ |
| Minibatch size | 32 |
| Q network: channels | 32, 64, 64 |
| Q-network: filter size | $8 \times 8, 4 \times 4, 3 \times 3$ |
| Q-network: stride | 4, 2, 1 |
| Q-network: hidden units | 512 |
| Initial collect steps | 1600 |
| $n$-step | 10 |
| Training iterations | 40 |
| Training environment steps per iteration | 10K |
| **ReDo Parameter** | **Value** |
| Recycling period | 1000 |
| $\tau$-Dormant | 0.025 |
| Minibatch size for estimating neurons score | 64 |
| **Adaptive RR Parameter** | **Value** |
| check interval | 2000 |
| threshold | 0.001 |
| low Replay Ratio | 0.5 |
| high Replay Ratio | 2 |

Table 5: **Evaluation of Sample Efficiency on Atari-100k.** We report the scores and the mean and median HNSs achieved by different methods on Atari-100k.

| Game | Human | Random | DrQ($\epsilon$) (RR=0.5) | DrQ($\epsilon$) (RR=1) | DrQ($\epsilon$) (RR=2) | ReDo (RR=1) | Adaptive RR (RR0.5to2) |
|---|---|---|---|---|---|---|---|
| Alien | 7127.7 | 227.8 | 815 | 865 | 917 | 794 | **935** |
| Amidar | 1719.5 | 5.8 | 114 | 138 | 133 | 163 | **200** |
| Assault | 742.0 | 222.4 | 755 | 580 | 579 | 675 | **823** |
| Asterix | 8503.3 | 210.0 | 470 | **764** | 442 | 684 | 519 |
| Bank Heist | 753.1 | 14.2 | 451 | 232 | 91 | 61 | **553** |
| Boxing | 12.1 | 0.1 | 16 | 9 | 6 | 9 | **18** |
| Breakout | 30.5 | 1.7 | 17 | **20** | 13 | 15 | 16 |
| Chopper Command | 7387.8 | 811.0 | 1037 | 845 | 1129 | **1650** | 1544 |
| Crazy Climber | 35829.4 | 10780.5 | 18108 | 21539 | 17193 | **24492** | 22986 |
| Demon Attack | 1971.0 | 152.1 | 1993 | 1321 | 1125 | 2091 | **2098** |
| Enduro | 861 | 0 | 128 | 223 | 138 | **224** | 200 |
| Freeway | 29.6 | 0.0 | 21 | 20 | 20 | 19 | **23** |
| Kung Fu Master | 22736.3 | 258.5 | 5342 | 11467 | 8423 | 11642 | **12195** |
| Pong | 14.6 | −20.7 | −16 | −10 | **3** | −6 | −9 |
| Road Runner | 7845.0 | 11.5 | 6478 | 11211 | 9430 | 8606 | **12424** |
| Seaquest | 42054.7 | 68.4 | 390 | 352 | 394 | 292 | **451** |
| SpaceInvaders | 1669 | 148 | 388 | 402 | 408 | 379 | **493** |
| Mean HNS (%) | 100 | 0 | 42.3 | 41.3 | 35.1 | 42.3 | **55.8** |
| Median HNS (%) | 100 | 0 | 22.6 | 30.3 | 26.0 | 41.6 | **48.7** |
| # Superhuman | N/A | 0 | 3 | 1 | 1 | 2 | **4** |
| # Best | N/A | 0 | 0 | 2 | 1 | 3 | **11** |

