# OpenReview forum: "Revisiting Plasticity in Visual Reinforcement Learning: Data, Modules and Training Stages"
_ICLR.cc/2024/Conference — ICLR 2024 poster_

### Official Review · Reviewer_c6tM · 2023-10-29

**Soundness:** 3 good
**Presentation:** 3 good
**Contribution:** 2 fair
**Rating:** 6
**Confidence:** 4

**Summary:**

This paper studies the plasticity loss problem in visual reinforcement learning. By using Fraction of Active Units (FAU) as a metric to measure the plasticity loss, several observations are concluded:
- Data augmentation is critical.
- The critic part in the actor-critic framework suffers from the plasticity loss most.
- Plasticity loss in early states is irrevocable.

The authors also propose a technique to mitigate the plasticity loss issue, by using adaptive replay ratio.  Experiments are mainly conducted across 6 DMC tasks, showing the improvements of the proposed technique.

**Strengths:**

- **Nice paper writing**. The motivation, experiments, and conclusions from observations are well stated, making this paper interesting to read.
- **Good ablations**. The careful experiments on different modules and stages are interesting, which could give some insights into Visual RL.
- **Interesting measurement**. The select measurement (FAU) is interesting. Though similar things have been well studied in pure deep RL [1,2], this work might be an early work to study the activation units in visual-based deep RL.

[1] Nikishin, Evgenii, et al. "The primacy bias in deep reinforcement learning." ICML 2022.

[2] Sokar, Ghada, et al. "The dormant neuron phenomenon in deep reinforcement learning." ICML 2023.

**Weaknesses:**

- **Limited insights revealed**. As the analysis in this work shows, the most critical factor for visual RL is **data augmentation (DA)**. However, this has been a very well-known factor for the community, and a quite large amount of recent works have extensively studied DA in visual RL. I think this work mainly gives a new perspective to explain the effectiveness of DA. This is certainly interesting, but considering the metric is directly from the previous works and similar measurements have been also extensively studied, applying FAU simply to a well-known critical factor does not give enough insights to the community.

- **Limited technical contributions**. The only technical contribution of this work is to apply a low replay ratio and a high replay ratio in different stages. This is simple yet not effective enough to support its simplicity and also is not very related to the entire story that this work tells.

- **Limited evaluations**. The experiments are only conducted on 6 DMC tasks, and therefore the diversity in domains and tasks are both limited. It could be good to see the observation and the technique are more universal, considering their simplicity.

Though I like the analysis given in this work, I think this paper is not qualified for the acceptance of ICLR, with the reasons above.

**Questions:**

See *weakness* above.

---

> ### Author Response · Authors · 2023-11-18
> **Response to the Weakness 1-1**
>
> Thank you for your feedback! Here are our responses:
>
> We acknowledge your perspective that data augmentation (DA) is a well-established factor in the field of Visual Reinforcement Learning (VRL). However, the relationship between DA and plasticity loss remains under-explored, which constitutes a primary contribution of our study. This aspect has also been favorably recognized by all the other three reviewers. Thus, we respectfully submit that our study provides more nuanced insights than a mere reaffirmation of DA's effectiveness. Specifically, our investigations yield substantial insights into at least the following three key areas:
>
> ---
>
> ### **1. For Data Augmentation:**
>
> 🌟 **Effectively mitigating catastrophic plasticity loss presents the most plausible explanation for the indispensability of data augmentation in achieving sample-efficient VRL, rather than merely offering a new perspective on the effectiveness of DA.**
>
> Contrary to its incremental role in enhancing test performance in supervised learning tasks such as image classification, **DA plays an indispensable and decisive role in numerous visual RL tasks**. As illustrated in [1], even simple random shift transformations applied to input observations can lead to significant performance improvements in previously unsuccessful algorithm. By analogy, if DA in supervised learning boosts algorithm performance from 80 to 90 points, in VRL it's more like transforming algorithms from a baseline of 20 points to an impressive 90.
>
> Despite the DA operations in VRL being similar to those in supervised CV tasks, the explanations for DA's underlying mechanisms in CV do not readily apply to VRL scenarios. For example, DA can be considered as an a priori mechanism in tasks with distribution shifts between training and testing phases, adeptly mitigating these discrepancies. However, [2] note that all the environments from DMC employ a camera that is fixed relative to the agent’s position. Hence, robustness to shifts does not appear to introduce any useful inductive bias about the underlying tasks. Additionally, previous studies have widely held that in supervised vision tasks, DA can enhance representation learning by inducing invariance or equivariance constraints [3]. However, evidence from [4] demonstrates that adding explicit alignment regularization to DA does not improve its performance, but rather, can have a detrimental effect.
>
> In summary, to the best of our knowledge, existing research fails to adequately explain how DA fundamentally impacts VRL training, turning previously ineffective algorithms into highly sample-efficient ones. Our study introduces a novel connection between DA and plasticity within VRL training, demonstrating that DA can significantly mitigate plasticity loss in the critic module. In the absence of DA, the detrimental impact of catastrophic plasticity loss hinders the effective training of VRL agents, thus elucidating the pronounced performance disparity in VRL scenarios with or without DA and providing a newperspective on the underlying working mechanism of DA.
>
> [1] Ma G, Zhang L, Wang H, et al. Learning Better with Less: Effective Augmentation for Sample-Efficient Visual Reinforcement Learning. NeurIPS 2023.
>
> [2] Cetin E, Ball P J, Roberts S, et al. Stabilizing off-policy deep reinforcement learning from pixels. ICML 2022.
>
> [3] Wang H, Huang Z, Wu X, et al. Toward learning robust and invariant representations with alignment regularization and data augmentation. KDD 2022
>
> [4] Klee D, Walters R, Platt R. Understanding the Mechanism behind Data Augmentation’s Success on Image-based RL. RLDM 2022.

---

> ### Author Response · Authors · 2023-11-18
> **Response to the Weakness 1-2**
>
> ### **2. For Visual RL Community:**
>
> 🌟 **We uncover that the primary obstacle to achieving sample-efficient VRL is, in fact, the catastrophic plasticity loss of the critic module, rather than the previously widely attributed poor visual representation capability of the encoder. Our detailed deconstruction of plasticity loss in VRL offers a clear and novel pathway for enhancing its sample efficiency.**
>
> Given that the primary distinction between VRL and state-based RL lies in the concurrent optimization of task-specific policies and learning of compact state representations from high-dimensional observations, previous studies have naturally attributed the sample inefficiency in VRL primarily to **inadequate visual representation capabilities**.Recent advancements in VRL, focusing on enhancing sample efficiency, have primarily centered around the development of advanced representation learning techniques to forge more effective encoders. However, recent empirical studies [5, 6] suggest that these methods do not consistently improve training efficiency, indicating that insufficient representation may not be the primary bottleneck hindering the sample efficiency of current algorithms.
>
> Although recent studies increasingly acknowledge that merely enhancing visual representation no longer yields significant improvements in sample efficiency, the primary impediment to further advancements in VRL sample efficiency remains elusive. This paper, through a deconstruction of plasticity loss in VRL, identifies that **the catastrophic plasticity loss in the critic module is indeed this key bottleneck**. As `reviewer Kom1` notes, 'the paper can provide a promising potential for a new methodology to deal with plasticity loss,' our research at least suggests two viable pathways for further enhancing VRL sample efficiency: 1. Designing more effective interventions to better maintain the critic's plasticity, with data augmentation (DA) being an example of such interventions; 2. Utilizing the critic's plasticity level as a metric to dynamically adjust the algorithmic design of RL training, as exemplified by Adaptive Reward Reshaping (ARR).
>
> [5] Li X, Shang J, Das S, et al. Does self-supervised learning really improve reinforcement learning from pixels?. NeurIPS 2022.
>
> [6] Hansen N, Yuan Z, Ze Y, et al. On Pre-Training for Visuo-Motor Control: Revisiting a Learning-from-Scratch Baseline. ICML 2023.

---

> > ### Author Response · Authors · 2023-11-18
> > **Response to the Weakness 1-3**
> >
> > ### **3. For Plasticity Loss in Deep RL:**
> >
> > *Plasticity loss is an intrinsic challenge in DRL agent training, essential to address for developing efficient DRL algorithms[7]. In this work, we conduct a systematic empirical exploration focusing on three primary underexplored facets and derive the following insightful conclusions:*
> >
> > 🌟**① We uniquely identify DA as a highly effective method for mitigating plasticity loss in VRL, a previously unaddressed aspect in plasticity research.  This highlights the critical role of data-centric approaches in preserving plasticity, which is under-explored in previous literatures.**
> >
> > > `Reviewer m8KG`: The paper offers a fresh take on plasticity -- I wouldn't have predicted that data augmentation would be such an effective regularizer, and this phenomenon is **not an obvious conclusion** from past literature.
> >
> > Previous methods focusing on maintaining DRL's plasticity have primarily centered on resetting network parameters (e.g., ReDo [8]), adding explicit regularization to network weights (such as InFeR [9]), or altering network architecture (like C-ReLU [10]). In contrast, data-centric intervention methods, exemplified by DA, have been significantly overlooked. As `Reviewer m8KG` notes, the potent role of DA in maintaining plasticity was not an obvious conclusion in past research. `Section 3` of our paper shows that DA outperforms other interventions in preventing catastrophic plasticity loss, underscoring the need for greater focus on data-centric methods in addressing these issues.
> >
> > 🌟**② We are the first to perform a structured analysis on the varied characteristics of plasticity loss in different modules and training stages, enhancing understanding of this pivotal issue in DRL and suggesting more precise strategies for its resolution.**
> >
> > Firstly, we establish that the critic's plasticity loss is catastrophic, while the actor can train effectively with a sparser state. This discovery not only provides vital guidelines for accurately measuring an agent's plasticity level, but also offers fresh insights into the **network architecture** design for actors and critics. For instance, our research reveals that the plasticity of the critic is crucial for training efficiency. In contrast, the lower plasticity level induced in the actor by data augmentation does not adversely affect training. This finding leads to a natural insight: an actor that is sparse or small in scale may suffice for efficient training. We conducted a preliminary experiment to test this hypothesis. In the original DrQ-v2 setup, both the actor and critic networks' hidden layers are set at a dimension of 1024. We experimented with doubling or halving the dimensions of either the actor or critic network. The results presented in the following table show that, contrary to the assertions in [11] regarding the expansion of the Q network's width enhancing sample efficiency in DQN-based methods, **altering the width of the actor within a certain range does not significantly impact training efficiency, while the method presents much more sensitive to the width of the critic.** This insight, emerging naturally from our findings, highlights the significance of our investigation into the varying plasticities of different modules and their impact on training, even as it awaits further validation.
> >
> > #### a. Finger Spin (after 100k steps)
> >
> > (AWS = Actor Width Scaling, CWS = Critic Width Scaling)
> >
> > |                 | Episode Return |   |                 | Episode Return |
> > |:---------------:|:--------------:|:-:|:---------------:|:--------------:|
> > | AWS=0.25, CWS=1 | $724.7\pm122$  |   | AWS=1, CWS=0.25 | $586.5\pm63$   |
> > | AWS=0.5, CWS=1  | $736.4\pm133$  |   | AWS=1, CWS=0.5  | $694.1\pm89$   |
> > | AWS=1, CWS=1    | $738.6\pm14$   |   | AWS=1, CWS=1    | $738.6\pm14$   |
> > | AWS=2, CWS=1    | $725.8\pm31$   |   | AWS=1, CWS=2    | $794.2\pm66$   |
> > | AWS=4, CWS=1    | $733.4\pm19$   |   | AWS=1, CWS=4    | $811.3\pm25$   |
> >
> > #### b. Cheetah Run (after 100k steps)
> >
> > |                 | Episode Return |   |                 | Episode Return |
> > |:---------------:|:--------------:|:-:|:---------------:|:--------------:|
> > | AWS=0.25, CWS=1 | $483.5\pm126$  |   | AWS=1, CWS=0.25 | $233.0\pm233$  |
> > | AWS=0.5, CWS=1  | $492.0\pm18$   |   | AWS=1, CWS=0.5  | $417.3\pm22$   |
> > | AWS=1, CWS=1    | $492.6\pm31$   |   | AWS=1, CWS=1    | $492.6\pm31$   |
> > | AWS=2, CWS=1    | $477.8\pm91$   |   | AWS=1, CWS=2    | $515.0\pm74$   |
> > | AWS=4, CWS=1    | $455.4\pm117$  |   | AWS=1, CWS=4    | $553.9\pm52$   |
> >
> > Additionally, we identify the importance of timely recovering the critic's plasticity in the early stages of training. As `Reviewer m8KG` aptly points out, 'Section 5, which identifies "critical periods" in the training process, is particularly insightful as it suggests that regularization can be modulated throughout training.'

---

> > > ### Author Response · Authors · 2023-11-18
> > >
> > > [7] Lyle C, Zheng Z, Nikishin E, et al. Understanding plasticity in neural networks. ICML 2023.
> > >
> > > [8] Sokar G, Agarwal R, Castro P S, et al. The dormant neuron phenomenon in deep reinforcement learning. ICML 2023.
> > >
> > > [9] Lyle C, Rowland M, Dabney W. Understanding and preventing capacity loss in reinforcement learning. ICLR 2022.
> > >
> > > [10] Abbas Z, Zhao R, Modayil J, et al. Loss of plasticity in continual deep reinforcement learning. Priprint 2023.
> > >
> > > [11] Schwarzer M, Ceron J S O, Courville A, et al. Bigger, Better, Faster: Human-level Atari with human-level efficiency. ICML 2023.

---

> ### Author Response · Authors · 2023-11-18
> **Response to the Weakness 2**
>
> We recognize the reviewer's concerns in Weakness 2, mainly regarding the effectiveness and applicability of Adaptive RR, and its relevance to the concepts revisited in earlier sections. In our subsequent response, we'll further validate Adaptive RR's efficacy and versatility. For now, our focus is on clarifying the deep-seated link between our methods and the extensive exploration we conducted on data, modules, and training stages.
>
> 🌟 **Our discovery that the critic’s plasticity is a pivotal factor in sample efficiency suggests that the FAU of the critic module could be adaptively used to discern the current training stage.** Without a thorough investigation into how the plasticity of different modules differently impacts training, it would be challenging to precisely select the critic’s FAU as a metric for dynamically guiding the adjustments in RR. In networks with an actor-critic architecture, it might seem intuitive to use the average FAU of both actor and critic as the metric for guiding RR adjustments. However, as illustrated in Figure 17 in our paper's appendix, the divergent trends of FAU in actor and critic can lead to confusion, thereby obscuring the observable trends in the agent's plasticity.
>
> 🌟 **Our investigation into the distinct properties of plasticity across different training stages directly motivates the implementation of Adaptive RR.** Initially, a low RR is employed to prevent catastrophic plasticity loss. As training progresses and plasticity dynamics stabilize, increasing the RR can enhance reuse frequency. While the method of gradually increasing RR is simple, its effectiveness stems from our nuanced understanding of the complex issue of plasticity loss in DRL. Furthermore, our investigation provides robust empirical support for all the implementation details of the Adaptive RR approach.
>
> 🌟 **The significant impact of DA on plasticity offers us a comparative experimental approach to study modules and stages.** Section 3 of our paper not only reveals DA as an effective intervention to mitigate plasticity loss, but also the pronounced plasticity differences caused by the presence or absence of DA provide a compelling basis for comparative analysis. This allows for a deeper exploration of how plasticity varies and evolves across different modules and training stages.

---

> ### Author Response · Authors · 2023-11-18
> **Response to the Weakness 3**
>
> ### **1. Effectiveness:**
>
> The emergence of the high RR dilemma stems from the fact that increasing the replay ratio can enhance sample efficiency by boosting data reuse frequency, yet this is often counteracted by the more severe plasticity loss that accompanies it. Adaptive RR effectively capitalizes on the sample efficiency benefits of a high RR, while skillfully circumventing the dire plasticity loss consequences.
>
> Prior to introducing Adaptive RR, the most effective method to address the high RR dilemma involved periodically resetting parts of the network while employing a high static RR. In the table below, we present a comparison between RR=2 (with Reset) and Adaptive RR across three representative tasks. Compared to high RR with Reset, Adaptive RR demonstrates superior advantages in at least three aspects:
>
> (1) **Higher sample efficiency.** While applying Resets to a high replay ratio of 2 does improve its sample efficiency, it does not match the performance levels attained by Adaptive RR.
>
> (2) **Greater computing efficiency.** Unlike high RR with Reset, which consistently maintains a high update frequency, Adaptive RR employs a lower RR in the initial stages of training, thereby enhancing computing efficiency.
>
> (3) **Enhanced safety during interaction.** As emphasized in [1], the performance collapse caused by Resets can lead to a higher likelihood of violating safety constraints during agent-environment interactions. Adaptive RR, on the other hand, effectively avoids this issue.
>
>
> |  Return after 2M Steps  |   | RR=0.5 (w/o Reset) |   | RR=2 (w/o Reset)|   |RR=2 (w/ Reset)|  | Adaptive RR |
> |:-------------|---|:------:|---|:----:|---|:------------:|---|:-----------:|
> | Cheetah Run   |   |$828\pm59$||$793\pm9$| |**$885\pm20$**|   |  $880\pm45$ |
> | Walker Run    |   |$710\pm39$||$709\pm7$| |  $749\pm10$  |  |**$758\pm12$**|
> | Quadruped Run |   |$579\pm120$||$417\pm110$||$511\pm47$  |  |**$784\pm53$**|
>
>
> [1] Kim W, Shin Y, Park J, et al. Sample-Efficient and Safe Deep Reinforcement Learning via Reset Deep Ensemble Agents. NeurIPS 2023.
>
> ### **Applicability:**
>
> Firstly, while our original manuscript primarily reported the experimental results from 6 DMC tasks, these tasks encompass a diverse range of challenging characteristics in **continuous control tasks**. For instance, `Finger Turn Hard` presents a typical sparse reward setting, while `Quadruped Run` features a large state-action space. The consistent performance improvement of our proposed Adaptive RR across these tasks demonstrates its broad applicability in diverse continuous control environments.
>
> Furthermore, we acknowledge the reviewer's concerns about Adaptive RR's universality. To validate its effectiveness across varied tasks, we extensively tested it in the **discrete Atari-100k tasks**. The ensuing data, as shown in the table below, demonstrate that Adaptive RR surpassed both lower and higher static RR settings in **15 of the 17 diverse tasks**. Conducted with 5 seeds per task, these findings will be elaborated in our forthcoming manuscript revision, highlighting Adaptive RR's broad applicability.
>
>
> | Task            |   | RR=0.5 |   | RR=2 |   | ARR (from 0.5 to 2) |
> |:----------------|---|:------:|---|:----:|---|:-------------------:|
> | Alien           |   |$815\pm133$||$917\pm132$||  **$935\pm94$**   |
> | Amidar          |   |$114\pm58$| |$133\pm57$| |  **$200\pm53$**   |
> | Assault         |   |$755\pm119$||$579\pm44$| |  **$823\pm94$**   |
> | Asterix         |   |$470\pm65$| |$442\pm69$| |  **$519\pm36$**   |
> | BankHeist       |   |$451\pm114$||$91\pm34$|  |  **$553\pm68$**   |
> | Boxing          |   |$16\pm9$|   |$6\pm2$|    |  **$18\pm5$**     |
> | Breakout        |   |**$17\pm6$**||$13\pm3$|  |    $16\pm5$       |
> | ChopperCommand  |   |$1073\pm318$||$1129\pm114$||**$1544\pm519$** |
> | CrazyClimber    |   |$18108\pm2336$||$17193\pm3883$||**$22986\pm2265$**|
> | DemonAttack     |   |$1993\pm678$||$1125\pm191$||**$2098\pm491$** |
> | Enduro          |   |$128\pm45$|  |$138\pm36$|  |**$200\pm32$**   |
> | Freeway         |   |$21\pm4$|    |$20\pm3$|    |**$23\pm2$**     |
> | KungFuMaster    |   |$5342\pm4521$||$8423\pm4794$||**$12195\pm5211$**|
> | Pong            |   |$-16\pm3$|   |**$3\pm10$**|   |  $-10\pm12$  |
> | RoadRunner      |   |$6478\pm5060$||$9430\pm3677$||**$12424\pm2826$**|
> | Seaquest        |   |$390\pm79$|  |$394\pm85$|  |**$451\pm69$**   |
> | SpaceInvaders   |   |$388\pm122$| |$408\pm93$|  |**$493\pm93$**   |

---

> ### Comment · Reviewer_c6tM · 2023-11-21
> **Thank the authors for the response**
>
> I acknowledge the efforts made by the authors. I think my major concern "limited insights revealed" is partially alleviated, regarding the new discussion. However,  I slightly disagree with the new emphasis from authors:
> > We uncover that the primary obstacle to achieving sample-efficient VRL is, in fact, the catastrophic plasticity loss of the critic module, rather than the previously widely attributed poor visual representation capability of the encoder. Our detailed deconstruction of plasticity loss in VRL offers a clear and novel pathway for enhancing its sample efficiency.
>
> I think the visual representation ability is still the primary obstacle. As cited by the authors, the paper [6] actually shows that data augmentation is the key factor not only for visual RL but also for other settings like imitation learning. I think the claim made by authors "the primary obstacle to achieving sample-efficient VRL is, in fact, the catastrophic plasticity loss of the critic module, rather than the previously widely attributed poor visual representation capability of the encoder" lacks enough support, since the critic part also shares the visual representation.
>
> Thank the authors for their detailed discussions and additional experiments and I would raise my score accordingly. Further considering the understanding from previous papers, I would tend to not give a positive score.
>
> [6] Hansen N, Yuan Z, Ze Y, et al. On Pre-Training for Visuo-Motor Control: Revisiting a Learning-from-Scratch Baseline. ICML 2023.

---

> > ### Author Response · Authors · 2023-11-21
> > **Clarification on the Relationship between Visual Representation and Plasticity Loss**
> >
> > We sincerely thank the reviewer for the precise pointers regarding our expressions and claims. We recognize that our previous statements could easily mislead readers and were not sufficiently rigorous. Following the reviewer's advice, we will rephrase our claims for greater accuracy and update our manuscript in subsequent versions. Here, we wish to present to the reviewer our revised claims about the impact of different modules on achieving sample-efficient VRL, and elucidate the rationale behind these conclusions.
> >
> > > **The poor visual representation capacity of the encoder is `not the only contributor` to sample inefficiency in VRL. Even with a superior visual representation, the catastrophic plasticity loss of the critic module remains a significant primary obstacle in achieving sample-efficient VRL.**
> >
> > We agree with the reviewer's observation that an encoder with superior visual representation capability is essential for sample-efficient and high-performance VRL agents.  This capability is fundamental for the actor and critic's decision-making. However, our paper makes a pivotal contribution by emphatically demonstrating that **while high-level visual representation is essential, it is not sufficient on its own to ensure sample-efficient VRL. The plasticity loss in the critic module still poses a substantial obstacle to achieving sample-efficient VRL.**
> >
> > We substantiate our conclusion through experiments with a 'frozen ImageNet pre-trained encoder.' Specifically, instead of training the encoder from scratch, we adopt the experimental setup described in [1], which employs an ImageNet pre-trained ResNet model as the agent’s encoder, keeping its parameters frozen throughout the training process. Building on this setup, we compare the effects of employing DA against not using it on sample efficiency, thereby isolating and negating the potential influences from disparities in the encoder’s representation capability on training. The training curves are illustrated in `Figure 4 of our paper` and the comparative results at various step checkpoints are reported in the table below. These results demonstrate that employing DA consistently outperforms scenarios without DA by a significant margin.
> >
> > |||w/o DA||w/ DA||Improvement|
> > |:-:|-|:--:|:-:|:-:|-|:-:|
> > | **Walker Stand (0.5M)** ||  448±148   |   |  798±152  |   |  **+78.0%**   |
> > | **Walker Stand (1M)**   ||  558±123   |   |  923±52   |   |  **+65.4%**   |
> > ||||||||
> > | **Walker Run (1M)**||232±64    |   |  621±26   |   |  **+167.5%**  |
> > | **Walker Run (2M)**||323±75    |   |  678±17   |   |  **+109.7%**  |
> > | **Walker Run (3M)**||  378±87    |   |  715±13   |   |  **+89.5%**   |
> >
> > 🌟 On one hand, the challenge in training sample-efficient VRL agents without DA, despite having a powerful encoder, indicates that **solely relying on a high-capacity encoder is insufficient for true sample efficiency in VRL**. 🌟 On the other hand, the introduction of DA leads to a substantial improvement in sample efficiency. This suggests that **other significant hurdles exist, hindering the effective training of both the actor and critic**, even with strong visual representation abilities. In later sections of our paper, we convincingly illustrate that **this major hurdle is the catastrophic plasticity loss of the critic module**. This is convincingly demonstrated using Plasticity Injection as a diagnostic tool, as shown in `Figure 5 of our paper`.
> >
> > It is important to note that our paper does not aim to diminish the importance of visual representation in VRL. Instead, we seek to highlight an equally crucial issue that demands the VRL community's attention: the catastrophic plasticity loss of critic. We acknowledge the need for more precise language in our previous statements and we plan to revise them in our subsequent manuscript as per the reviewer's suggestions.
> >
> > Indeed, we posit that our exploration of a bottleneck beyond representation in VRL holds significant implications for the community. Over recent years, the focus of the VRL community has predominantly been on enhancing representation learning through advanced designs, such as incorporating self-supervised auxiliary tasks or integrating pre-trained encoders. However, recent studies [2] [3] suggest that these approaches may no longer yield substantial improvements in sample efficiency on existing algorithms. In contrast, concurrent research, such as DrM [4], has empirically demonstrated that mitigating plasticity loss can remarkably enhance the sample efficiency of VRL. This indicates that **maintaining plasticity is a promising avenue for achieving more powerful and efficient VRL**. Our paper provides numerous insights into understanding this issue in depth.
> >
> > Thanks again for the valuable suggestions, which are helpful in enhancing the quality of our manuscript. We look forward to further discussions with the reviewer.

---

> > > ### Author Response · Authors · 2023-11-21
> > > **Reference**
> > >
> > > [1] Yuan Z, Xue Z, Yuan B, et al. Pre-trained image encoder for generalizable visual reinforcement learning. NeurIPS 2022.
> > >
> > > [2] Hansen N, Yuan Z, Ze Y, et al. On Pre-Training for Visuo-Motor Control: Revisiting a Learning-from-Scratch Baseline. ICML 2023.
> > >
> > > [3] Li X, Shang J, Das S, et al. Does self-supervised learning really improve reinforcement learning from pixels? NeurIPS 2022.
> > >
> > > [4] Xu G, Zheng R, Liang Y, et al. DrM: Mastering Visual Reinforcement Learning through Dormant Ratio Minimization. Preprint.

---

> > > ### Comment · Reviewer_c6tM · 2023-11-23
> > > **Thank you for your response**
> > >
> > > Thank the authors for their further discussions.
> > >
> > > I appreciate the overall analysis given in the paper, which has been emphasized in my primary review. My major concern about this paper is still its actual contribution to the Visual RL community. All complex analysis finally still points out that, data augmentation, more specifically, random shift, is the critical factor.
> > >
> > > I personally like the analysis from this paper. However, it has also been stated in my primary review,
> > > > As the analysis in this work shows, the most critical factor for visual RL is data augmentation (DA). However, this has been a very well-known factor for the community, and a quite large amount of recent works have extensively studied DA in visual RL. I think this work mainly gives a new perspective to explain the effectiveness of DA.
> > >
> > > Considering the effectiveness of DA has been analyzed very sufficiently in previous works [1,2,3], utilizing the tools from [4,5,6,7] to analyze this factor again is not interesting enough to me. The only extra contribution from the experiments is that DA affects the critic most, instead of the actor part. This certainly reveals some insights but is not interesting enough for acceptance.
> > >
> > > **This paper might be interesting for people who are familiar with plasticity loss since they might be unfamiliar with the common sense that DA is the critical part,  but not interesting for people familiar with visual RL due to the context.**
> > >
> > > Again I appreciate the efforts made by authors. The storytelling and analysis in this paper are still worth reading for the community. I would maintain my score considering the reasons above however.
> > >
> > >
> > > [1] Ma, Guozheng, et al. "Learning Better with Less: Effective Augmentation for Sample-Efficient Visual Reinforcement Learning." arXiv preprint arXiv:2305.16379 (2023).
> > >
> > > [2] Li, Lu, et al. "Normalization Enhances Generalization in Visual Reinforcement Learning." arXiv preprint arXiv:2306.00656 (2023).
> > >
> > > [3] Hansen, Nicklas, et al. "On Pre-Training for Visuo-Motor Control: Revisiting a Learning-from-Scratch Baseline." arXiv preprint arXiv:2212.05749 (2022).
> > >
> > > [4] Lyle C, Zheng Z, Nikishin E, et al. Understanding plasticity in neural networks. ICML 2023.
> > >
> > > [5] Sokar G, Agarwal R, Castro P S, et al. The dormant neuron phenomenon in deep reinforcement learning. ICML 2023.
> > >
> > > [6] Lyle C, Rowland M, Dabney W. Understanding and preventing capacity loss in reinforcement learning. ICLR 2022.
> > >
> > > [7] Abbas Z, Zhao R, Modayil J, et al. Loss of plasticity in continual deep reinforcement learning. Priprint 2023.

---

> > > > ### Author Response · Authors · 2023-11-23
> > > > **Our Contributions to the VRL Community**
> > > >
> > > > We sincerely appreciate the reviewer's response and **their recognition of our contribution to the study of plasticity loss**. However, we wish to re-emphasize to the reviewer the broader significance and contribution of this paper to the Visual RL community, which extends well beyond merely 'providing a new perspective on the effectiveness of DA'.
> > > >
> > > > ### 1. **We highlight that poor visual representation capacity is `not the only contributor` to sample inefficiency in VRL, and compellingly demonstrate that catastrophic plasticity loss of critic is `an equally significant bottleneck`.**
> > > >
> > > >
> > > > Over recent years, the VRL community has mainly concentrated on enhancing representation learning by implementing advanced strategies, including self-supervised auxiliary tasks and the integration of pre-trained encoders. However, contemporary studies [1,2] indicate that such methods might not significantly boost sample efficiency in existing algorithms anymore. Contrarily, parallel research like [DrM](https://arxiv.org/abs/2310.19668) [3] has empirically shown that mitigating plasticity loss can significantly improve the sample efficiency of VRL. This suggests that focusing on **maintaining plasticity could be a promising approach to develop more potent and efficient VRL systems**. Our paper provides numerous insights into understanding this issue in depth.
> > > >
> > > >
> > > > ### 2. **DA, though beneficial, is not sufficient to maintain plasticity in scenarios with high RR, which have potential to further improve sample efficiency. Our proposed Adaptive RR strategy effectively balances this data reuse frequency with plasticity loss, thereby improving sample efficiency.**
> > > >
> > > > In Section 3 of our paper, we underscored the potent effectiveness of DA in preserving plasticity under default configurations with a RR of 0.5. However, while DA stands as one of the most effective interventions, it may still fall short in maintaining plasticity under high RR settings. Following Reviewer m8KG's suggestion, we conducted supplementary experiments of ReDO [4], a method effectively restoring agent plasticity through resetting neurons. The results of these comparisons are presented in the following table. At a lower RR of 0.5, introducing ReDO does not enhance sample efficiency, indicating that DA alone is sufficient to maintain plasticity in this scenario. However, at a higher RR of 2, incorporating ReDO significantly improves training outcomes, suggesting that even with DA, catastrophic plasticity loss can occur at high RR scenarios. Thus, **`DA plays a critical role, but it alone is not sufficient to fully address the challenges in VRL.`**
> > > >
> > > > Furthermore, as shown in the following table, Adaptive RR consistently outperforms static RR configurations across three challenging tasks, even when the latter incorporates ReDO. This not only demonstrates that Adaptive RR can effectively balances between data reuse frequency and plasticity loss, but also shows that dynamically adjusting the RR based on the critic's overall plasticity level enables competitive performance through effective neuron-level network parameter resetting.
> > > >
> > > >
> > > > |2M Env Steps | |RR=0.5 (default)| RR=0.5 (w/ ReDO) | | RR=2 (default) | RR=2 (w/ ReDO) | | ARR (from 0.5 to 2) |
> > > > |:------------|-|:--------------:|:----------------:|-|:--------------:|:--------------:|-|:-------------------:|
> > > > |Cheetah Run  | | 828 $\pm$ 59   |   788 $\pm$ 5    | |  793 $\pm$ 9   |  873 $\pm$ 19  | |  **880 $\pm$ 45**   |
> > > > |Walker Run   | | 710 $\pm$ 39   |   618 $\pm$ 50   | |  709 $\pm$ 7   |  734 $\pm$ 16  | |  **758 $\pm$ 12**   |
> > > > |Quadruped Run| | 579 $\pm$ 120  |   371 $\pm$ 158  | |  417 $\pm$ 110 |  608 $\pm$ 53  | |  **784 $\pm$ 53**   |
> > > >
> > > >
> > > > [1] Hansen N, Yuan Z, Ze Y, et al. On Pre-Training for Visuo-Motor Control: Revisiting a Learning-from-Scratch Baseline. ICML 2023.
> > > >
> > > > [2] Li X, Shang J, Das S, et al. Does self-supervised learning really improve reinforcement learning from pixels? NeurIPS 2022.
> > > >
> > > > [3] Xu G, Zheng R, Liang Y, et al. DrM: Mastering Visual Reinforcement Learning through Dormant Ratio Minimization. arXiv preprint arXiv:2310.19668, 2023.
> > > >
> > > > [4] Sokar G, Agarwal R, Castro P S, et al. The dormant neuron phenomenon in deep reinforcement learning. ICML 2023.
> > > >
> > > > ---
> > > >
> > > > Thank you once again for your valuable feedback. Your insights and discussions are instrumental in refining our manuscript. We hope that our responses have sufficiently addressed your concerns we are looking forward to further discussion on any remaining question about our paper and response.

---

> ### Comment · Reviewer_c6tM · 2023-11-23
> **Thank you for bringing me the attention to the potential of plasticity loss**
>
> It is interesting to see that another recent work DrM explores a similar thing and shows the potential of mitigating plasticity loss. I would be glad to see a direct discussion and comparison with DrM in the paper, such as the main difference, deeper connection, and further potential. **I would consider raising my score accordingly.**
>
> Also, I hope all the related papers that authors mention during the discussion period should be included in their main paper.

---

> > ### Author Response · Authors · 2023-11-23
> >
> > Thank you for your response. Next, we will briefly summarize the comparison between DrM and our paper.
> >
> > **Summary of DrM**: DrM uses three core mechanisms to guide agents’ exploration-exploitation trade-offs by actively minimizing the dormant ratio: 1. Dormant-ratio-guided perturbation; 2. Awaken exploration scheduler; and 3. Dormant-ratio-guided exploitation.
> >
> >
> > **Deeper Connection**:
> > 1. Both our research and the DrM underscore the significance of plasticity in the learning process of VRL.
> > 2. Our Adaptive RR and DrM share a similarity in their approach: both use the agent's plasticity level as a key metric to dynamically adjust certain parameters of RL training. While DrM focuses on dynamically altering the balance between exploration and exploitation, our Adaptive RR specifically targets the dynamic adjustment of the replay ratio. This commonality highlights the significance of plasticity level as a crucial metric for optimizing RL training strategies.
> >
> > **Main Difference**:
> > 1. DrM is focused on achieving more effective exploration-exploitation trade-offs, while Adaptive RR is designed to find a superior balance between data reuse frequency and plasticity loss.
> > 2. Our paper delves deeply into the investigation of plasticity loss in VRL, offering valuable insights for the community to further address this issue.
> >
> >
> > **Further Potential**:
> > We argue that Adaptive RR and DrM target different aspects of the learning process: while DrM adjust the balance between exploration and exploitation, our method adapt replay ratio. Combining these approaches, in our view, leads to improved sample efficiency by leveraging the strengths of both.
> >
> >
> > Once DrM releases their code, we will conduct further experiments to compare the combined effects of both methods and update the findings in our subsequent manuscript, as well as the in-depth discussion.

---

> ### Comment · Reviewer_c6tM · 2023-11-23
> **Score raising**
>
> Thank you!
>
> It would be good to see these discussions in the updated manuscript in the future and more in-depth experiments.
>
> I have raised my score.

---

> > ### Author Response · Authors · 2023-11-23
> >
> > Thank you for your precious time on the review and the appreciation on our work. Your suggestions are crucial for enhancing the quality of our manuscript, and we will incorporate your recommendations into subsequent revisions.

---

### Official Review · Reviewer_m8KG · 2023-10-30

**Soundness:** 3 good
**Presentation:** 4 excellent
**Contribution:** 3 good
**Rating:** 6
**Confidence:** 4

**Summary:**

This paper studies the interaction between data augmentation, replay ratios, and dormant neurons in deep reinforcement learning. In particular, it demonstrates a striking similarity in the efficacy of data augmentation and resets in preserving plasticity in deep RL agents, and shows that data augmentation mitigates a decline in the fraction of active units in the network. It further shows that the fraction of active units can also be used to design an adaptive replay ratio scheme which tailors the replay ratio to the phase of learning in the network.

**Strengths:**

- The paper is an enjoyable read. It is very well-written and presented; claims and evidence are clear and easy to follow, and the figures are clearly explained and easy to interpret
  - The paper offers a fresh take on plasticity -- I wouldn't have predicted that data augmentation would be such an effective regularizer, and this phenomenon is not an obvious conclusion from past literature.
  - The study of manipulation tasks presents a nice counterpoint to prior work on plasticity which has focused largely on Atari games.
  - I appreciated the comparisons against a variety of interventions in Figure 2, highlighting the effect size of data augmentation in contrast to other regularizers that had been studied in the literature.
  - The work is able to ground its evaluation of plasticity not only in the performance of the agents on the RL tasks, but also using the Fraction of Active Units metric, which gives some insight into what is going on inside the network. This is useful because performance-only metrics are unable to disentangle plasticity from other factors in the RL training process such as exploration and generalization.
  - The idea of adapting the replay ratio based on the stability of the network's training process makes a lot of sense and, despite the potential issues I highlight below with this particular instantiation, could be a generally useful quantity to keep track of.
  - Section 5, which identifies "critical periods" in the training process, is particularly insightful as it suggests that regularization can be modulated throughout training

**Weaknesses:**

- The paper highlights the FAU as a major indicator of plasticity in the network, but glaringly omits evaluations of ReDO as a baseline.
    - The domains studied in this paper are all from similar environments, and I'm not sure how much this paper is telling us about general properties of data augmentation, replay ratios, and plasticity, vs how much it is telling us about the interaction between these things in a specific class of robotic manipulation tasks.
    - The causality around FAU and plasticity is quite vague in this paper. I wasn't sure whether it was claiming that the FAU is a *symptom* of plasticity loss, and that DA and ARR are affecting some hidden causal factor which reduces both plasticity loss and increases FAU, or whether the claim was that *by maintaining the FAU* these interventions are able to avoid plasticity loss.
    - Data augmentation has the opposite on FAU in the actor and the critic, but it is not explained why a higher FAU would be beneficial in the actor but not the critic. In particular, data augmentation *dramatically* reduces the FAU in the actor, without seeming to have any ill effects. This seems to suggest that FAU might not be a particularly useful measure of plasticity for all learning problems, or at least that the story isn't as simple as that described in the paper.
    - The specific mechanism by which data augmentation is helping to prevent plasticity loss is unclear. While it is observed to correlate with a reduction in the number of dormant neurons, it isn't obvious whether this is a causal mechanism I would like to see an evaluation of a few data augmentation classes on at least one environment to see how the choice of data augmentation influences its effect on plasticity.
    - Minor nit: the claim that using a fixed encoder network means that the effect of plasticity on the representation is completely eliminated is too strong, as there is still some representation learning happening in the actor and critic MLPs. This should be noted in the paper text.

**Questions:**

- The definition of FAU is a bit unclear: is it measuring average activity of neurons over a large batch, i.e. the sparsity of the network's representation, or is the expression inside the indicator function nonzero if the unit is active for *any* input, i.e. FAU =  1 - (fraction of dormant neurons)?
- The replay ratio adaptation method is not described in sufficient detail. How specifically is the "recovery of plasticity" measured? How robust is the method to different variations on this measurement?

---

> ### Comment · Reviewer_m8KG · 2023-11-19
>
> I appreciate that the authors have prioritized responding to low-scoring reviewers first, but I would like to emphasize that I think this paper, with a few changes, could merit a >6 score, as I think Figure 1 alone is especially striking and worth being highlighted in a conference paper. Specifically:
>
> 1. The paper really needs to evaluate against ReDO. This will also help a lot to clarify the somewhat murky causality around dormant neurons and learning dynamics.
>
> 2. Including even a single non-Mujoco domain (ideally a more discrete, visually rich setting like an environment from Atari-100K or ProcGen) would go a long way towards validating the generality of the observations in this paper. It is worth noting that [Schwarzer et al.](https://arxiv.org/pdf/2305.19452.pdf) noticed significant benefits from incorporating resets, though they evaluate on an expensive domain. Perhaps MinAtar could be a cheaper alternative?
>
> 3. A more mechanistic characterization of the effect of data augmentation (in lieu of additional evaluation environments) would give some indication of how environment-specific this strategy is. For example, to what extent is data augmentation mitigating overfitting/generalization issues vs network optimization pathologies?

---

> > ### Author Response · Authors · 2023-11-19
> >
> > Many thanks to the reviewer's active involvement and support of our work. We are conducting further experiments based on your feedback to confirm our findings and the effectiveness of our method. We apologize for our late response, due to needing more experiments and limited computing resources.
> >
> > > Next, we will provide updates on the progress and findings from the completed experiments that align with the reviewer's key concerns. Regarding the ongoing experimental validations, we are actively advancing these and will post the results as soon as they are completed.
> >
> > ---
> >
> > ### **1. ReDO**
> >
> > We fully agree with the reviewer that ReDO should be considered as a baseline for comparison. ReDO achieves a balance between recovering an agent's plasticity and avoiding performance collapse by precisely identifying and resetting dormant neurons. ReDO exemplifies an approach that mitigates plasticity loss by directly addressing it within the network's architecture. Orthogonal to the ReDO's approach, Adaptive RR dynamically adjusts the replay ratio based on the critic's FAU as a metric for the agent's plasticity level, aiming to find a more optimal tradeoff. Comparing ReDO and Adaptive RR will indeed advance our understanding of plasticity loss. In fact, **we're currently replicating ReDO for this comparative analysis. We plan to finish these experiments soon and look forward to further discussions with the reviewer.**
> >
> > ---
> >
> > ### **2. Atari-100K**
> >
> > Aligning with the reviewer's latest suggestion, we also recognize the importance and necessity of extending our analysis and evaluations to discrete control tasks, complementing our work on continuous DMC tasks. We have initiated experiments in Atari-100k, using DrQ($\epsilon$) as our baseline, which is based on DQN.
> >
> > Firstly, we have indeed observed that the Q-network in DQN-based architectures exhibits  the same FAU trend to the critic network in actor-critic architectures. Specifically, it initially declines rapidly, then begins to rise, then begins to rise, and finally stabilizes at a higher level before gradually decreasing again. The analysis of our Atari-100k experiments will be further refined and updated in subsequent versions of our manuscript.
> >
> > Furthermore, our proposed **Adaptive RR surpassed both lower and higher static RR settings in 15 of the 17 Atari tasks**. Each task underwent five random runs. These results strongly demonstrate the universality of Adaptive RR.
> >
> >
> > | Task            |   | RR=0.5 |   | RR=2 |   | ARR (from 0.5 to 2) |
> > |:----------------|---|:------:|---|:----:|---|:-------------------:|
> > | Alien           |   |$815\pm133$||$917\pm132$||  **$935\pm94$**   |
> > | Amidar          |   |$114\pm58$| |$133\pm57$| |  **$200\pm53$**   |
> > | Assault         |   |$755\pm119$||$579\pm44$| |  **$823\pm94$**   |
> > | Asterix         |   |$470\pm65$| |$442\pm69$| |  **$519\pm36$**   |
> > | BankHeist       |   |$451\pm114$||$91\pm34$|  |  **$553\pm68$**   |
> > | Boxing          |   |$16\pm9$|   |$6\pm2$|    |  **$18\pm5$**     |
> > | Breakout        |   |**$17\pm6$**||$13\pm3$|  |    $16\pm5$       |
> > | ChopperCommand  |   |$1073\pm318$||$1129\pm114$||**$1544\pm519$** |
> > | CrazyClimber    |   |$18108\pm2336$||$17193\pm3883$||**$22986\pm2265$**|
> > | DemonAttack     |   |$1993\pm678$||$1125\pm191$||**$2098\pm491$** |
> > | Enduro          |   |$128\pm45$|  |$138\pm36$|  |**$200\pm32$**   |
> > | Freeway         |   |$21\pm4$|    |$20\pm3$|    |**$23\pm2$**     |
> > | KungFuMaster    |   |$5342\pm4521$||$8423\pm4794$||**$12195\pm5211$**|
> > | Pong            |   |$-16\pm3$|   |**$3\pm10$**|   |  $-10\pm12$  |
> > | RoadRunner      |   |$6478\pm5060$||$9430\pm3677$||**$12424\pm2826$**|
> > | Seaquest        |   |$390\pm79$|  |$394\pm85$|  |**$451\pm69$**   |
> > | SpaceInvaders   |   |$388\pm122$| |$408\pm93$|  |**$493\pm93$**   |

---

> > > ### Author Response · Authors · 2023-11-19
> > >
> > > ### **3. Opposing FAU Trends of Actor and Critic**
> > >
> > > The distinct effects of DA on Actor and Critic networks are indeed intriguing and merit further investigation. In Section 4, using Plasticity Injection [1] as a diagnostic tool, we showed that catastrophic plasticity loss in the Critic network hinders efficient training. However, the reason why DA significantly reduces the FAU in the Actor without adverse effects remains an open question for future research. Our current thinking on this issue is twofold:
> > >
> > > 1. **As the reviewer rightly points out, FAU cannot be equated directly with a network's plasticity loss; it's more of a measure or dimension of plasticity level.** To more definitively quantify plasticity loss, we are exploring the inclusion of additional metrics such as feature rank and weight norm. We will report these analyses to the reviewer as soon as the experiments are completed.
> > >
> > > 2. **An actor that is sparse or small in scale may suffice for efficient training.** We conducted a preliminary experiment to test this hypothesis. In the original DrQ-v2 setup, both the actor and critic networks' hidden layers are set at a dimension of 1024. We experimented with doubling or halving the dimensions of either the actor or critic network. The results presented in the following table show that, contrary to the assertions in [2] regarding the expansion of the Q network's width enhancing sample efficiency in DQN-based methods, **altering the width of the actor within a certain range does not significantly impact training efficiency, while the method presents much more sensitive to the width of the critic.** This insight, emerging naturally from our findings, highlights the significance of our investigation into the varying plasticities of different modules and their impact on training, even as it awaits further validation.
> > >
> > >     a. **Finger Spin (after 100k steps)**
> > >
> > >     (AWS = Actor Width Scaling, CWS = Critic Width Scaling)
> > >
> > >     |                 | Episode Return |   |                 | Episode Return |
> > >     |:---------------:|:--------------:|:-:|:---------------:|:--------------:|
> > >     | AWS=0.25, CWS=1 | $724.7\pm122$  |   | AWS=1, CWS=0.25 | $586.5\pm63$   |
> > >     | AWS=0.5, CWS=1  | $736.4\pm133$  |   | AWS=1, CWS=0.5  | $694.1\pm89$   |
> > >     | AWS=1, CWS=1    | $738.6\pm14$   |   | AWS=1, CWS=1    | $738.6\pm14$   |
> > >     | AWS=2, CWS=1    | $725.8\pm31$   |   | AWS=1, CWS=2    | $794.2\pm66$   |
> > >     | AWS=4, CWS=1    | $733.4\pm19$   |   | AWS=1, CWS=4    | $811.3\pm25$   |
> > >
> > >     **b. Cheetah Run (after 100k steps)**
> > >
> > >     |                 | Episode Return |   |                 | Episode Return |
> > >     |:---------------:|:--------------:|:-:|:---------------:|:--------------:|
> > >     | AWS=0.25, CWS=1 | $483.5\pm126$  |   | AWS=1, CWS=0.25 | $233.0\pm233$  |
> > >     | AWS=0.5, CWS=1  | $492.0\pm18$   |   | AWS=1, CWS=0.5  | $417.3\pm22$   |
> > >     | AWS=1, CWS=1    | $492.6\pm31$   |   | AWS=1, CWS=1    | $492.6\pm31$   |
> > >     | AWS=2, CWS=1    | $477.8\pm91$   |   | AWS=1, CWS=2    | $515.0\pm74$   |
> > >     | AWS=4, CWS=1    | $455.4\pm117$  |   | AWS=1, CWS=4    | $553.9\pm52$   |
> > >
> > >
> > > [1] Nikishin E, Oh J, Ostrovski G, et al. Deep Reinforcement Learning with Plasticity Injection. NeurIPS 2023.
> > >
> > > [2] Schwarzer M, Ceron J S O, Courville A, et al. Bigger, Better, Faster: Human-level Atari with human-level efficiency. ICML 2023.
> > >
> > > ---
> > >
> > > ### **4. Data Augmentation**
> > >
> > > We are currently employing various metrics, including FAU, weight norm, feature rank and Hessian's rank, to assess the differing impacts of various DA types on plasticity. The results of these experiments will be discussed with the reviewer as soon as they are completed.
> > >
> > > In fact, in VRL tasks, one of the most commonly used DA operation that significantly enhances sample efficiency involves padding the original observation from $84 \times 84$ to $92 \times 92$, followed by a random crop back to $84 \times 84$. This augmentation method can be considered as a form of random minor perturbation. As noted in [3], all environments from the DMC suite use a camera fixed relative to the agent's position, indicating that robustness to shifts may not provide any useful inductive bias regarding the underlying tasks. Thus, the DA discussed in our paper is likely not aimed at narrowing the generalization gap. Instead, it appears to affect network dynamics to maintain plasticity.
> > >
> > > [3] Cetin E, Ball P J, Roberts S, et al. Stabilizing off-policy deep reinforcement learning from pixels. ICML 2022.
> > >
> > > ---
> > >
> > > > **Key ongoing experiments planned for updates:**
> > > > 1. ReDO;
> > > > 2. Additional metrics to quantify the plasticity of actor and critic;
> > > > 3. Evaluation on different DA classes and their influence on plasticity.

---

> > > > ### Comment · Reviewer_m8KG · 2023-11-19
> > > >
> > > > Thanks to the authors for the follow-up. The Atari-100K results look promising, but to properly evaluate them it would be useful to see what DrQ($\epsilon$) scores with the default RR used by Kostrikov et al. (for example, if the best RR value is 1 then the adaptive approach might outperform the suboptimal fixed values of 0.5 and 2 but underperform the optimal fixed value). The comparison of actor and critic architectures is also interesting, and is consistent with the results of [Raileanu et al.](https://proceedings.mlr.press/v139/raileanu21a/raileanu21a.pdf), [Lyle et al.](https://arxiv.org/abs/2206.02126), and [Cobbe et al.](https://arxiv.org/abs/2009.04416), which merit discussion as related work.
> > > >
> > > > I look forward to seeing the ReDO results.

---

> > > > > ### Author Response · Authors · 2023-11-20
> > > > > **ReDO in Atari**
> > > > >
> > > > > We are deeply thankful for the reviewer's valuable feedback, which has significantly contributed to the enhancement of our manuscript. The references you mentioned will be discussed in our forthcoming revision. We have completed **our comparison with ReDO in Atari tasks**, which demonstrates that Adaptive RR continues to exhibit strong competitiveness. Additionally, our latest discovery reveals that **DA effectively elevates the feature rank of the actor network**, providing us with a deeper understanding of the actor's plasticity.
> > > > >
> > > > > ---
> > > > >
> > > > > ### **1. ReDO and Optimal Static `RR=1`**
> > > > >
> > > > > We fully agree with the reviewer's suggestion that a comparison with DrQ($\epsilon$) using the default RR is essential. Such a comparison is crucial to highlight the necessity and superiority of dynamically adjusting the RR based on the critic's plasticity. Due to time and computational resource constraints, we have adopted the default RR results from all tasks except SpaceInvaders and Enduro directly from the BBF’s paper [1]. Additionally, **we evaluated ReDO's performance in the following 17 tasks using its original source code**, with each task undergoing 5 random runs. The complete results from our Atari-100k experiments are presented in the table below:
> > > > >
> > > > > 🌟 **Adaptive RR outperforms other configurations, including ReDO, in 11 out of the 17 tasks.** This not only demonstrates that Adaptive RR can strike a superior trade-off between reuse frequency and plasticity loss, but also shows that dynamically adjusting the RR based on the agent's overall plasticity level (specifically referring to the Q network or critic) enables competitive performance through effective neuron-level network parameter resetting.
> > > > >
> > > > > > Please note that **we are still diligently working to replicate ReDO in the DMC tasks**. Given that the original ReDO study did not include experiments in the DMC environment, additional time is required to refine the experimental details, such as parameter tuning, to effectively implement ReDO in this context. We will engage in further discussions with the reviewer as soon as there are significant developments in our experiments.
> > > > >
> > > > >
> > > > > |Task|RR=0.5|RR=2|ARR (from 0.5 to 2)|RR=1 (optimal static RR)|ReDO (RR=1)|
> > > > > |:---:|:---:|:---:|:---:|:---:|:---:|
> > > > > |Alien|815±133|917±132|**935±94**|865.2|794±47|
> > > > > |Amidar|114±58|133±57|**200±53**|137.8|163±56|
> > > > > |Assault|755±119|579±44|**823±94**|579.6|675±57|
> > > > > |Asterix|470±65|442±69|519±36|**763.6**|684±98|
> > > > > |BankHeist|451±114|91±34|**553±68**|232.9|61±43|
> > > > > |Boxing|16±9|6±2|**18±5**|9.0|9±5|
> > > > > |Breakout|17±6|13±3|16±5|**19.8**|15±6|
> > > > > |ChopperCommand|1073±318|1129±114|1544±519|844.6|**1650±343**|
> > > > > |CrazyClimber|18108±2336|17193±3883|22986±2265|21539.0|**24492±4641**|
> > > > > |DemonAttack|1993±678|1125±191|**2098±491**|1321.5|**2091±588**|
> > > > > |Enduro|128±45|138±36|200±32|**223.5**|**224±35**|
> > > > > |Freeway|21±4|20±3|**23±2**|20.3|19±11|
> > > > > |KungFuMaster|5342±4521|8423±4794|**12195±5211**|11467.4|11642±5459|
> > > > > |Pong|-16±3|**3±10**|-10±12|-9.1|-6±14|
> > > > > |RoadRunner|6478±5060|9430±3677|**12424±2826**|11211.4|8606±4341|
> > > > > |Seaquest|390±79|394±85|**451±69**|352.3|292±68|
> > > > > |SpaceInvaders|388±122|408±93|**493±93**|402.1|379±87|
> > > > >
> > > > > [1] Schwarzer M, Ceron J S O, Courville A, et al. Bigger, Better, Faster: Human-level Atari with human-level efficiency.ICML 2023.
> > > > >
> > > > > ---
> > > > >
> > > > > ### **2. DA can maintain a higher feature rank in actor.**
> > > > >
> > > > > We further explored the use of additional metrics to observe the plasticity of different modules, including Weight Norm and [Feature Rank](https://github.com/google-research/google-research/tree/master/generalization_representations_rl_aistats22/coherence). The results of these experiments have been updated in Figure 19 of the paper's appendix.
> > > > >
> > > > > The results indicate that **introducing DA significantly enhances the feature rank of the actor**. Combined with previous observations that DA results in a low FAU for the actor network, we can infer that **a sparse actor network can possess the necessary expressivity**. Conversely, the critic must maintain sufficient plasticity and active states to continually learn the non-stationary target from newly collected experiences [2].
> > > > >
> > > > >
> > > > > [2] Implicit Under-Parameterization Inhibits Data-Efficient Deep Reinforcement Learning. ICLR 2021.

---

> > > > > ### Author Response · Authors · 2023-11-22
> > > > > **ReDO in DMC**
> > > > >
> > > > > We appreciate the reviewer's continued interest and engagement. Our experiments implementing ReDO in DMC tasks have now been completed, with results presented in the following table. Compared to the DQN-based ReDO in Atari tasks, the DDPG-based ReDO we replicated in DMC tasks has the following main differences in implementation details:
> > > > >
> > > > >
> > > > > 1. **We only apply the ReDO operation to neurons within the critic network.** Recognizing that critic's plasticity loss significantly impacts training, our implementation of ReDO in DMC tasks is tailored to reset neurons only in the critic module. We recognize the importance of conducting a thorough ablation study to evaluate the impact of applying ReDO across different modules. However, given our current time constraints, a detailed exploration of this aspect is planned for future manuscript revisions.
> > > > > 2. **We use a threshold $\tau=0.025$ and frequency $F=2\times10^5$ in our experiments.** In our preliminary experiments, a threshold value of $\tau=0.025$ proved more effective than $\tau=0.1$. The latter caused greater fluctuations in training, which was detrimental to efficient training.
> > > > >
> > > > >
> > > > >
> > > > > |2M Env Steps | |RR=0.5 (default)| RR=0.5 (w/ ReDO) | | RR=2 (default) | RR=2 (w/ ReDO) | | ARR (from 0.5 to 2) |
> > > > > |:------------|-|:--------------:|:----------------:|-|:--------------:|:--------------:|-|:-------------------:|
> > > > > |Cheetah Run  | | 828 $\pm$ 59   |   788 $\pm$ 5    | |  793 $\pm$ 9   |  873 $\pm$ 19  | |  **880 $\pm$ 45**   |
> > > > > |Walker Run   | | 710 $\pm$ 39   |   618 $\pm$ 50   | |  709 $\pm$ 7   |  734 $\pm$ 16  | |  **758 $\pm$ 12**   |
> > > > > |Quadruped Run| | 579 $\pm$ 120  |   371 $\pm$ 158  | |  417 $\pm$ 110 |  608 $\pm$ 53  | |  **784 $\pm$ 53**   |
> > > > >
> > > > >
> > > > > The performance comparison between ReDO under different static RR and our proposed *Adaptive RR* offers the following insights:
> > > > >
> > > > > 🌟 **In high RR settings, ReDO significantly boosts sample efficiency.** This demonstrates that ReDO's method of precisely identifying and resetting dormant neurons is effective in mitigating plasticity loss and enhancing training efficiency. However, **in low RR scenarios, incorporating ReDO actually hinders performance**, implying that existing algorithms using DA don't suffer from catastrophic plasticity loss at these settings. This phenomenon aligns with ReDO's performance in Atari tasks, as illustrated in `Figure 10` of [1].
> > > > >
> > > > > 🌟 **Adaptive RR consistently outperforms static RR in three challenging tasks, even when the latter incorporates ReDO.** This not only demonstrates that Adaptive RR can effectively balances between data reuse frequency and plasticity loss, but also shows that dynamically adjusting the RR based on the critic's overall plasticity level enables competitive performance through effective neuron-level network parameter resetting.
> > > > >
> > > > >
> > > > > [1] Sokar G, Agarwal R, Castro P S, et al. The dormant neuron phenomenon in deep reinforcement learning. ICML 2023.
> > > > >
> > > > > ---
> > > > >
> > > > > We are profoundly grateful for the reviewer's recognition, encouragement, and reminders. Your suggestions are incredibly insightful and crucial for refining our manuscript. The points discussed with you will be incorporated into subsequent revisions. We look forward to continuing our discussions with you.

---

> ### Author Response · Authors · 2023-11-20
> **Response to the Questions:**
>
> Here we provide clarifications and explanations in response to the questions raised by the reviewer.
>
> ---
>
> **Q1:** **The definition of FAU** is a bit unclear: is it measuring average activity of neurons over a large batch, i.e. the sparsity of the network's representation, or is the expression inside the indicator function nonzero if the unit is active for any input, i.e. FAU = 1 - (fraction of dormant neurons)?
>
> **A:** In our paper, the Fraction of Active Units (FAU) is defined as a metric for measuring the average activity of neurons over a large batch, in line with the approaches previously outlined in [1] and [2].  Thank you for your attention to this detail. We will further clarify this concept in the subsequent versions of our manuscript.
>
> [1] Lyle C, Zheng Z, Nikishin E, et al. Understanding plasticity in neural networks. ICML 2023.
>
> [2] Lee H, Cho H, Kim H, et al. PLASTIC: Improving Input and Label Plasticity for Sample Efficient Reinforcement Learning. NeurIPS 2023.
>
> ---
>
> **Q2:** The replay ratio adaptation method is not described in sufficient detail. How specifically is the **"recovery of plasticity"** measured? How robust is the method to different variations on this measurement?
>
> **A:** After initiating training, we monitor the FAU of the critic module at every check interval of $I$ steps. When the FAU difference between consecutive checkpoints falls below the minimal threshold $\tau$, it signifies the **'recovery of plasticity'**. In both DMC and Atari environments, the minimal threshold $\tau$ is set to $0.001$; for DMC, the interval $I$ is set to $5 \times 10^4$ steps, and for Atari, it is set to $2 \times 10^3$ steps. This experimental design stems from the observation that the critic's FAU rapidly decreases during the early stages of training and subsequently recovers. Once its value stabilizes at a relatively high level, the trend in FAU change becomes more gradual. Consequently, a minimal FAU difference between successive checkpoints suggests that the agent has moved beyond the early stages, which are more susceptible to catastrophic plasticity loss. At this point, we can safely increase the replay ratio to pursue higher sample efficiency.
>
> Furthermore, compared to judging the critic's FAU against a specific absolute value, our method of observing FAU trends to determine its training stage offers greater robustness. As illustrated in Figure 17 of our Appendix, setting a specific FAU value as the criterion for 'recovery' often requires adjusting the threshold for each specific task. In contrast, our approach demonstrates considerable robustness in hyperparameters. The minimal threshold $\tau$, set at $0.001$ for both DMC and Atari environments, effectively works across various tasks. The only adjustment needed is selecting an appropriate check interval based on the total allowed steps.

---

### Official Review · Reviewer_Kom1 · 2023-11-01

**Soundness:** 3 good
**Presentation:** 3 good
**Contribution:** 3 good
**Rating:** 6
**Confidence:** 4

**Summary:**

This paper addresses the issue of plasticity loss where RL agents struggle to learn from new experiences after being trained on non-stationary data. The authors suggest three key findings. Firstly, they discovered that data augmentation (DA) holds greater significance in alleviating plasticity loss compared to reset. Secondly, their extensive experiments demonstrate that the bottleneck for training lies in the plasticity loss of the critic module. Lastly, restoring the critic's plasticity to an adequate level in the early stages is an important factor. They analyzed plasticity of the neural network for the static high Replay Ratio (RR) during RL agent training and based on their findings, they introduced an adaptive replay ratio which shows superior performance compared to its static counterpart and effectively addresses the high RR dilemma.

**Strengths:**

-	The paper is well-written and easy to follow.

-	Their suggested method, adaptive RR, is straightforward and intuitive. The results demonstrate promising potential for the proposed approach.

-	Well-established experiments demonstrated the effectiveness of DA and the importance of Critic's plasticity.

**Weaknesses:**

-	Their proposed methodology, adaptive RR, is based on the analysis of the correlation between replay ratio and plasticity, which is relatively unrelated to the DA. This makes main message of this paper unclear. Are the authors emphasizing the importance of DA or are the authors emphasizing the importance of adaptive RR? The manuscript would benefit if it were to make their main message clearer by aligning their findings and their proposed method. It seems that although the findings regarding the relationship between DA and reset is intriguing, the main method is unrelated with their explanation regarding the effectiveness of DA.

-	Their experiments on data augmentation w/ and w/o reset contradict existing research. According to Nikishin et. al. [1], they conducted a comparison between DrQ w/ reset and DrQ w/o reset, finding that the DrQ w/ reset demonstrated superiority over DrQ w/o reset (Check Figure 18 in [1]). Perhaps the results driven by the author are driven by the high variance of data augmentation w/o reset as in [1] or due to the difference in the DMC environments selected for evaluation. I have intentions to raise my score if the authors are able to show experimental results that their findings regarding the relationship between DA & reset also holds for other DMC environments.

-	In addition, the current replay ratio values utilized for adaptive RR seems to be limited to a small range (e.g., [0.5, 1, 2]). In order to properly evaluate the efficacy of the proposed adaptive RR method, the authors should conduct further experiments for RR=4 and RR=8. Although higher RR does require longer training time and can be computationally expensive, it is essential for the authors to provide the following results in order to verify whether the proposed adaptive RR is indeed superior compared to static RR. It will also be best if the authors are able to provide results for high RR & reset & DA to further validate their findings that DA is indeed better off along and not paired with reset.

-	As the authors note as a limitation, their experimental environment is confined to DMC, and they only demonstrate the effectiveness of Adaptive RR under basic configurations. However, it appears that the paper can provide a promising potential for a new methodology to deal with plasticity loss.

[1] The Primacy Bias in Deep Reinforcement Learning, Nikshin et. al., ICML 2022

**Questions:**

None

---

> ### Author Response · Authors · 2023-11-19
> **Response to the Weakness 1**
>
> We appreciate your thorough review and insightful feedback. We will address each of your comments and concerns below and also in our revised manuscript.
>
> ---
>
> **W1:** The manuscript would benefit if it were to make their main message clearer by **aligning their findings and their proposed method**.
>
> **A:** We are grateful to the reviewer for their attention to the structure of our paper. Indeed, the development of Adaptive RR is intricately linked to our in-depth analysis of plasticity loss in VRL.
>
> The challenge of the high RR dilemma in DRL is well-recognized, stemming from a trade-off between improved sample efficiency due to greater data reuse and the accompanying catastrophic plasticity loss. Our thorough examination of plasticity from various perspectives provided a clear path to address this issue. Every part of our paper connects together to make a complete story. Without all these parts, we wouldn’t have been able to suggest Adaptive RR as a simple but effective way to deal with the high RR problem.
>
> 🌟 **Our discovery that the critic’s plasticity is a pivotal factor in sample efficiency suggests that the FAU of the critic module could be adaptively used to discern the current training stage.** Without a thorough investigation into how the plasticity of different modules differently impacts training, it would be challenging to precisely select the critic’s FAU as a metric for dynamically guiding the adjustments in RR. In networks with an actor-critic architecture, it might seem intuitive to use the average FAU of both actor and critic as the metric for guiding RR adjustments. However, as illustrated in Figure 17 in our paper's appendix, the divergent trends of FAU in actor and critic can lead to confusion, thereby obscuring the observable trends in the agent's plasticity.
>
> 🌟 **Our investigation into the distinct properties of plasticity across different training stages directly motivates the implementation of Adaptive RR.** Initially, a low RR is employed to prevent catastrophic plasticity loss. As training progresses and plasticity dynamics stabilize, increasing the RR can enhance reuse frequency. While the method of gradually increasing RR is simple, its effectiveness stems from our nuanced understanding of the complex issue of plasticity loss in DRL. Furthermore, our investigation provides robust empirical support for all the implementation details of the Adaptive RR approach.
>
> 🌟 **The significant impact of DA on plasticity offers us a comparative experimental approach to study modules and stages.** Section 3 of our paper not only reveals DA as an effective intervention to mitigate plasticity loss, but also the pronounced plasticity differences caused by the presence or absence of DA provide a compelling basis for comparative analysis. This allows for a deeper exploration of how plasticity varies and evolves across different modules and training stages.

---

> ### Author Response · Authors · 2023-11-19
> **Response to the Weakness 2**
>
> **W2:** ... I have intentions to raise my score if the authors are able to show experimental results that their findings regarding **the relationship between DA & reset** also holds for other DMC environments.
>
> **A:** Thank you for your interest in exploring whether incorporating Resets into DA can further enhance sample efficiency. Indeed, beyond the three tasks discussed in the main text, we have presented comparative results for eight other distinct tasks in Figures 11 and 12 of the Appendix. These tasks represent a diverse range of challenges and types within the DMC environments. Among these, only in 'Reacher Hard' did applying Resets noticeably improve the algorithm's sample efficiency and convergence performance.
>
> We also acknowledge the differences in our experimental results compared to the original study on Resets in [1]. We believe the primary reason for these discrepancies lies in our use of DrQ-v2[2] as the baseline algorithm, as opposed to the original DrQ[3] used in [1]. 🌟 On one hand, **DrQ-v2, with its sophisticated design enhancements, significantly improves sample efficiency and can effectively train in tasks where the original DrQ struggles.** For instance, Figure 18 in [1] indicates that DrQ fails to train effectively in tasks like Quadruped Walk, Quadruped Run, and Reacher Hard, where the introduction of Resets greatly enhances performance. In contrast, DrQ-v2 with DA already achieves high sample efficiency in these tasks, leaving little room for further improvement through Resets. 🌟 On the other hand, **DrQ sets the critic's replay ratio (RR) at 1, unlike the basic configuration of DrQ-v2, which is RR=0.5.** As discussed in [1], higher RR leads to more severe plasticity loss, making the improvements from Resets more pronounced. In the basic configuration of DrQ-v2 (RR=0.5), our experiments indeed show that DA alone is sufficient to maintain plasticity.
>
> However, as RR increases, the utility of Resets becomes apparent. We conducted comparative experiments in three tasks with and without Resets at RR=2. The results in the table below consistently show that Resets can further enhance sample efficiency by recovering plasticity in high RR scenarios, aligning with the conclusions in [1].
>
>
> |Task (after 2M Steps)||RR=2 (w/DA, w/o Reset)||RR=2 (w/ DA, w/ Reset)|
> |:-------------|---|:----:|---|:------------:|
> | Cheetah Run   ||$793\pm9$| |  $885\pm20$  |
> | Walker Run    ||$709\pm7$| |  $749\pm10$  |
> | Quadruped Run ||$417\pm110$||$511\pm47$  |
>
> [1] The Primacy Bias in Deep Reinforcement Learning, Nikshin et. al., ICML 2022
>
> [2] Yarats D, Fergus R, Lazaric A, et al. Mastering visual continuous control: Improved data-augmented reinforcement learning. ICLR 2022.
>
> [3] Denis Yarats, Ilya Kostrikov, and Rob Fergus. Image augmentation is all you need: Regularizing deep reinforcement learning from pixels. ICLR 2021.

---

> ### Author Response · Authors · 2023-11-19
> **Response to the Weakness 3 and Weakness 4**
>
> **W3:** ... it is essential for the authors to provide the following results in order to verify whether the proposed **adaptive RR is indeed superior compared to static RR**.
>
> **A:** To demonstrate the scalability of Adaptive RR, we evaluated its effectiveness under configurations of RR=4 and RR=8. As shown in the table below, 🌟 the performance of Adaptive RR (from 0.5 to 8) significantly exceed that of a static RR set at 8. 🌟 Furthermore, **Adaptive RR (from 0.5 to 4) not only surpass static high or low RR but also outperform the original ARR (from 0.5 to 2) in 2 out of 3 tasks** presented in the main text. This strongly validates the universality of Adaptive RR and its potential to achieve even greater efficacy.
>
>
>
> |Task (after 2M Steps)|   | RR=0.5 |   | RR=4 ||ARR (from 0.5 to 4) ||orignal ARR (0.5 to 2)|
> |:----------------|---|:------:|---|:----:|---|:-------------------:|---|:-----------------:|
> | Cheetah Run     |   |$828\pm59$ ||$802\pm7$ ||    $856\pm51$      ||  **$880\pm45$**  |
> | Walker Run      |   |$710\pm39$ ||$690\pm14$||  **$773\pm22$**    ||  $758\pm12$      |
> | Quadruped Run   |   |$579\pm120$||$297\pm88$||  **$799\pm42$**    ||  $784\pm53$      |
>
> |Task (after 2M Steps)|   | RR=0.5 |   | RR=8 ||ARR (from 0.5 to 8) ||orignal ARR (0.5 to 2)|
> |:----------------|---|:------:|---|:----:|---|:-------------------:|---|:---------------:|
> | Quadruped Run   |   |$579\pm120$||$162\pm85$||  **$439\pm86$**    ||  **$784\pm53$**    |
>
> ---
>
> **W4:** ...their experimental environment is confined to DMC...
>
> **A:** Firstly, while our original manuscript focused on 6 DMC tasks, these encompass a wide spectrum of challenging features within **continuous control tasks**. For example, `Finger Turn Hard` is characterized by a typical sparse reward environment, whereas `Quadruped Run` has an extensive state-action space. The consistent enhancement in performance with our proposed Adaptive RR across these tasks underscores its wide-ranging suitability for various continuous control scenarios. Additionally, as we have demonstrated in our `response to W3`, Adaptive RR has the potential to scale to even higher RR values.
>
> Furthermore, in response to the reviewer's concerns regarding the universality of Adaptive RR, we conducted comprehensive evaluations on Atari-100k tasks.  The results, as detailed in the forthcoming table, indicate that Adaptive RR consistently outperforms both lower and higher static RR **in 15 of 17 discrete control tasks**. Each task underwent five random runs, and these robust findings, which will be expanded upon in our revised manuscript, strongly validate Adaptive RR's broad applicability and effectiveness.
>
>
> | Task            |   | RR=0.5 |   | RR=2 |   | ARR (from 0.5 to 2) |
> |:----------------|---|:------:|---|:----:|---|:-------------------:|
> | Alien           |   |$815\pm133$||$917\pm132$||  **$935\pm94$**   |
> | Amidar          |   |$114\pm58$| |$133\pm57$| |  **$200\pm53$**   |
> | Assault         |   |$755\pm119$||$579\pm44$| |  **$823\pm94$**   |
> | Asterix         |   |$470\pm65$| |$442\pm69$| |  **$519\pm36$**   |
> | BankHeist       |   |$451\pm114$||$91\pm34$|  |  **$553\pm68$**   |
> | Boxing          |   |$16\pm9$|   |$6\pm2$|    |  **$18\pm5$**     |
> | Breakout        |   |**$17\pm6$**||$13\pm3$|  |    $16\pm5$       |
> | ChopperCommand  |   |$1073\pm318$||$1129\pm114$||**$1544\pm519$** |
> | CrazyClimber    |   |$18108\pm2336$||$17193\pm3883$||**$22986\pm2265$**|
> | DemonAttack     |   |$1993\pm678$||$1125\pm191$||**$2098\pm491$** |
> | Enduro          |   |$128\pm45$|  |$138\pm36$|  |**$200\pm32$**   |
> | Freeway         |   |$21\pm4$|    |$20\pm3$|    |**$23\pm2$**     |
> | KungFuMaster    |   |$5342\pm4521$||$8423\pm4794$||**$12195\pm5211$**|
> | Pong            |   |$-16\pm3$|   |**$3\pm10$**|   |  $-10\pm12$  |
> | RoadRunner      |   |$6478\pm5060$||$9430\pm3677$||**$12424\pm2826$**|
> | Seaquest        |   |$390\pm79$|  |$394\pm85$|  |**$451\pm69$**   |
> | SpaceInvaders   |   |$388\pm122$| |$408\pm93$|  |**$493\pm93$**   |

---

> ### Comment · Reviewer_Kom1 · 2023-11-22
>
> First of all, I thank the authors for their extensive and insightful discussion and would like to say that most of my concerns have been resolved. However, still I have some concerns regarding the author's comments and would like to leave a few comments below.
>
> Weakness 1 : Although I agree that the thorough investigation of whether FAU of the critic module or actor module is more important and investigating how adaptive RR operates is insightful, this still doesn't seem to be related to DA. How about removing the DA section from the manuscript? Although I understand that DA is an effective approach in mitigating plasticity loss, this still doesn't seem to be related with the major findings of this manuscript.
>
> Weakness 2 : Although it is interesting that Reset works differently between DrQ and DrQ-v2, your argument that DrQ's replay ratio is higher than of DrQ-v2 seems to be wrong. Check Appendix B.3 of [1] and Appendix B of [2]. From what I understand, DrQ seems to be using an update frequency of 1 while DrQ-v2 is using an update frequency of 2, which means that from the default setting DrQ-v2's replay ratio is higher, and naturally DrQ-v2 should enjoy Reset better than DrQ does (as shown by the extra experiments for Cheetah, Walker and Quadruped Run for RR=2). So, this raises doubt regarding the rationale of the phenomenon observed in the author's manuscript.
>
> [1] Image Augmentation is All You Need: Regularizing Deep Reinforcement Learning from Pixels, Kostrikov et. al., ICLR 2021
>
> [2] Mastering Visual Continuous Control: Improved Data-Augmented Reinforcement Learning, Yarats et. al., arXiv 2021
>
> Despite my remaining concerns, I personally found this manuscript insightful and enjoyed reviewing this paper. Therefore, I wish to increase my score to 6. Thank you for your hard work.

---

> > ### Author Response · Authors · 2023-11-22
> > **Response to the Weakness 1**
> >
> > We sincerely appreciate the reviewer's response and the recognition of our manuscript. Here, we seek to elucidate the **fundamental relationship between DA and the systematic framework underpinning our paper**. Furthermore, we will explicitly clarify that **the default RR in DrQ-v2 is indeed 0.5**, to address any misunderstandings between us and the reviewer.
> >
> > ---
> >
> > > **W1:** Although I do understand that DA is an effective approach in mitigating plasticity loss, this still does not seem to be related with the major findings of this manuscript.
> >
> > **A:** We acknowledge the reviewer's concern regarding the apparent lack of direct connection between our investigation of DA and the methods we ultimately propose. However, we wish to clarify that **our thorough demonstration of DA's effectiveness in mitigating plasticity loss provides an indispensable experimental tool for our subsequent analysis of modules and training stages**. Specifically, our initial insights were derived by observing the differences in the trends of Fraction of Active Units (FAU) across various modules at different stages, with and without the use of DA. These observations were then further substantiated through additional experiments. Thus, identifying the role of DA in maintaining plasticity is a critical component of the narrative structure of our paper. **Without this crucial aspect, the overall logical consistency of our manuscript would be significantly undermined.**

---

> ### Author Response · Authors · 2023-11-22
> **Response to the Weakness 2: Default RR in DrQ-v2**
>
> > **W2:** DrQ seems to be using an update frequency of 1 while DrQ-v2 is using an update frequency of 2, which means that from the default setting DrQ-v2's replay ratio is higher, and naturally DrQ-v2 should enjoy Reset better than DrQ does.
>
> **A:** We agree with the reviewer that the default RR in DrQ = 1.
>
> > DrQ: We perform one training update every time we receive a new observation.
>
> **However, we emphatically assert that the default RR for DrQ-v2 is indeed set at 0.5.**
>
> In light of the reviewer's comments, we revisited the 'default set of hyper-parameters' in the DrQ-v2 original paper's appendix and identified a potential source of the misunderstanding: **`Agent update frequency = 2`**. The naming of this hyperparameter could easily lead to confusion with the Replay Ratio (RR). However, in this context, `frequency=2` actually denotes that the agent is updated every two steps. Conversely, RR is defined as the number of gradient updates per environment step. **Hence, the default RR in DrQ-v2 should indeed be calculated as `1 / Agent update frequency`, equating to 0.5.**
>
> We further checked the [DrQ-v2 code](https://github.com/facebookresearch/drqv2) and found that `Agent update frequency = 2` is named as `update_every_steps: 2` in the code, as seen in [config.yaml#L41](https://github.com/facebookresearch/drqv2/blob/main/cfgs/config.yaml#L41). The code utilizing `update_every_steps` to adjust the update frequency appears in [drqv2.py#L230-L234](https://github.com/facebookresearch/drqv2/blob/main/drqv2.py#L230-L234). Specifically, it is implemented as follows:
> ```python
> def update(self, replay_iter, step):
>     metrics = dict()
>
>     if step % self.update_every_steps != 0:
>         return metrics
> ```
> This section of the code refers to updating the agent when `step` is an integer multiple of `update_every_steps`. The default value for `update_every_steps` is set to 2, thereby resulting in a default replay ratio of 0.5.
>
> It's also worth noting that the adjustment of the update frequency in the original implementation does not directly allow for a RR greater than 1. For example, setting `update_every_steps` to 0.5 still results in updating the agent every step, which effectively means an RR of 1.
>
> To accommodate higher RR settings in our experiments, we introduced an additional hyperparameter `replay_ratio`. We modified the code in [train.py#L180-L182](https://github.com/facebookresearch/drqv2/blob/main/train.py#L180-L182) to implement these higher RR settings. Our code modifications, including the introduction of the `replay_ratio` hyperparameter and adjustments to accommodate higher RR, are detailed in the `Supplementary Material`.
>
> ```python
> # try to update the agent (original)
> if not seed_until_step(self.global_step):
>     metrics = self.agent.update(self.replay_iter, self.global_step)
>     self.logger.log_metrics(metrics, self.global_frame, ty='train')
> ```
> >Our Revision:
> >
> >When RR>1:  set update_every_steps=1, self.agent.replay_ratio=RR;
> >
> >When RR<=1:  set update_every_steps=1/RR, self.agent.replay_ratio=1;
> ```python
> # try to update the agent (ours)
> if not seed_until_step(self.global_step):
>     for _iter in range(self.agent.replay_ratio):
>         metrics = self.agent.update(self.replay_iter, self.global_step)
>     self.logger.log_metrics(metrics, self.global_frame, ty='train')
> ```
>
> In conclusion, **the default RR for DrQ-v2 is indeed set at 0.5**. Furthermore, as demonstrated in `Figure 7` of our paper, the default RR of 0.5 achieves a high level of sample efficiency compared to settings of 1, 2, and 4. This optimal efficiency is attained with the lowest computational cost. Thus, **using the default setting as a high-level baseline, it is entirely reasonable for us to compare the effectiveness of DA and Reset in mitigating plasticity loss under these conditions**.
>
> Despite this, we greatly appreciate the reviewer's suggestion to compare DA and Reset under high RR conditions, which is crucial for enhancing our manuscript. Our experiments under high RR conditions are in complete agreement with the findings in [1]. We will add these comparisons to our forthcoming revision for a more thorough and meticulous examination of DA and Reset.
>
> [1] The Primacy Bias in Deep Reinforcement Learning, Nikshin et. al., ICML 2022.
>
> ---
>
> Thanks again for the valuable suggestions. We look forward to further discussions with the reviewer.

---

### Official Review · Reviewer_7W3D · 2023-11-02

**Soundness:** 3 good
**Presentation:** 3 good
**Contribution:** 3 good
**Rating:** 6
**Confidence:** 5

**Summary:**

This paper scrutinizes plasticity loss in visual reinforcement learning (VRL) by exploring data augmentation (DA), agent modules, and training stages. It reveals that DA is pivotal in mitigating plasticity loss, particularly within the critic module of VRL agents, which is identified as the main contributor to the loss of plasticity. To this end, the paper proposes an Adaptive Replay Ratio (RR) strategy that adjusts the RR based on the critic module's plasticity, significantly enhancing sample efficiency over static RR methods, as validated on the DeepMind Control suite.

**Strengths:**

This paper offers a novel perspective on the loss of plasticity in a visual reinforcement learning setup which includes:

-  **Analysis of Data Augmentation:**  A significant strength of this paper is its thorough analysis of why data augmentation is so effective in reinforcement learning. By conducting extensive experiments and analysis, the paper sheds light on the underlying mechanisms through which data augmentation enhances plasticity, particularly in visual reinforcement learning settings. This nuanced understanding helps bridge the gap between empirical success and theoretical underpinnings, providing valuable insights for future algorithm design.
-  **Module-Specific Analysis of Plasticity Loss:** Another strength is the paper's detailed dissection of plasticity loss across different agent modules, namely the encoder, actor, and critic. The authors' methodical comparison uncovers that the critic module is the primary culprit behind plasticity loss, challenging previous assumptions about the encoder's role.
- **A simple method to tackle the plasticity loss:** By proposing a simple, adaptive method to modulate the update ratio based on plasticity metrics, the paper presents a novel approach that improves upon the static replay ratio. This innovation has the potential to enhance the sample efficiency of diverse reinforcement learning models by aligning training intensity with the model's current capacity to learn.

**Weaknesses:**

This paper offers interesting insights and experimental results within the domain of visual reinforcement learning (VRL). Despite this, there are aspects that call into question the robustness of the findings:

- **Metrics-Related Concerns:** The primary metric used to assess plasticity loss, the fraction of active units, might be inadequate. The reliance on this metric is questionable since different activation functions such as CReLU and LeakyReLU can inherently bias the number of active units. In addition, as noted by Lyle et al. [1], fraction of active units does not fully encapsulate plasticity dynamics. I recommend that the authors broaden their analysis with additional metrics—like feature rank, weight norms, or even the curvature of the loss surface (e.g., the Hessian's rank)—to substantiate their claims more convincingly. While no single metric may definitively quantify plasticity loss, a consistency of findings across multiple measures would strengthen the empirical foundation of their arguments.

- **Analytical Concerns:** The paper claims the indispensability of data augmentation (DA) in mitigating plasticity loss, but it lacks a comprehensive analysis of why DA is superior. Other methods like layer normalization may have similar effects; what happens when these are combined, as done by Lee et al.? The paper would benefit from deeper exploration into why DA outperforms other methods. Providing a theoretical framework or empirical evidence for DA's efficacy in reducing plasticity loss would be a valuable addition to the field, potentially guiding future development of more robust VRL systems.

- **Applicability Concerns:**  I recognize that calling for additional experiments might seem an unfair request, as it can be levied against any paper. However, I do believe that extending the empirical validation to include at least one more analysis would greatly benefit the paper's credibility. It would be enlightening to see how the findings translate to non-actor-critic methods or across diverse benchmarks. For instance, assessing the proposed adaptive replay ratio method on a well-established sample-efficiency benchmark such as Atari-100k, with Rainbow algorithm, would improve the practical significance of the adaptive RR method. Without such additional validation, the scope and applicability of the conclusions remain in question.

I'm willing to increase the score if the authors address the outlined limitations.

[1] Understanding plasticity in neural networks., Lyle et al., ICML 2023.

[2] PLASTIC: Improving Input and Label Plasticity for Sample Efficient Reinforcement Learning., Lee et al., NeurIPS 2023.

**Questions:**

N/A

---

> ### Author Response · Authors · 2023-11-21
> **Response to the Weakness 1-2**
>
> We are grateful for your comprehensive review and valuable feedback. We apologize for the delayed response, as considerable time was required to complete the necessary experiments. We will address each of your comments and concerns in the following responses, as well as in our revised manuscript.
>
> ---
>
> **W1:** **Metrics-Related Concerns**: ... While no single metric may definitively quantify plasticity loss, a consistency of findings across **multiple measures** would strengthen the empirical foundation of their arguments.
>
> **A:** We fully concur with the reviewer's point that introducing additional metrics is essential for a more comprehensive characterization of an agent's plasticity. Beyond FAU, we have employed Weight Norm and Feature Rank as indicators to quantify plasticity. Specifically, we utilized these more comprehensive metrics to more accurately compare the characteristics of different modules and to contrast the effects of data augmentation with other interventions. The results of these experiments have been updated in `Figures 19 and 20 of the paper's appendix`. This revisitation of VRL's plasticity using a broader range of metrics has provided us with deeper insights into the distinct characteristics of the actor and critic.
>
> 🌟 DA significantly enhances the feature rank of both the actor and critic networks, effectively preventing implicit under-parameterization [1]. Combined with previous observations that DA results in a low FAU for the actor network and a high FAU for the critic network, we can infer that **a sparse actor network still retains the necessary expressivity**. Conversely, the **critic is required to maintain ample plasticity and active states to continuously learn from the non-stationary targets presented by newly collected experiences.**
>
> [1] Implicit Under-Parameterization Inhibits Data-Efficient Deep Reinforcement Learning. ICLR 2021.
>
> ---
>
> **W2:** **Analytical Concerns**: ... what happens when these are combined ... Providing a theoretical framework or empirical evidence for DA's efficacy in reducing plasticity loss would be a valuable addition to the field, potentially guiding future development of more robust VRL systems.
>
> **A:** The reviewer's suggestion for a more in-depth analysis and comparison of DA with other interventions is indeed necessary. It not only aids in understanding the powerful ability of DA to mitigate plasticity loss but also assists in designing more efficient VRL systems. As demonstrated in the PLASTIC study [2], combining various effective interventions can consistently maintain plasticity across a range of complex scenarios. Due to time and computational resource constraints, we plan to conduct experiments in future manuscript versions comparing DA with other interventions when combined, as [2] has already adequately demonstrated the potential of such combinations. During this rebuttal period, we will focus our discussion on **why DA, as opposed to interventions like layer normalization, possesses a stronger capability in maintaining plasticity**. To gain a deeper understanding, we employed a variety of metrics to measure the differences in training dynamics when using various interventions. The experimental results are displayed in `Figures 20 of the paper's appendix` and will be incorporated into the main text in subsequent versions of the manuscript. Our principal findings and understandings can be summarized as follows:
>
> 🌟 Although CReLU, by altering the activation function, forces the FAU of the critic to always be 0.5, it does not significantly enhance the critic's feature rank. This implies that while this method ensures each neuron is activated, these neurons do not possess sufficient expressivity.
>
> 🌟 Layer Normalization (LN) fails to effectively recover the FAU of the critic in the early stages of training. This inability prevents it from continuously learning from new experiences in later stages, aligning with the trend observed in its episode return training curve. After initial effective learning, algorithms using only LN quickly reach a training plateau, maintaining suboptimal performance without further improvement.
>
> 🌟 Employing DA effectively elevates the critic's FAU to a high level in the initial training phase, thereby avoiding catastrophic plasticity loss. This allows the agent to continuously optimize the accuracy of Q estimation through interactions with the environment. Additionally, with DA, we also achieve an actor with high expressivity. These two aspects together contribute to the effectiveness of DA in maintaining plasticity, making it an essential method for achieving sample-efficient VRL.
>
> [2] PLASTIC: Improving Input and Label Plasticity for Sample Efficient Reinforcement Learning. NeurIPS 2023.

---

> ### Author Response · Authors · 2023-11-21
> **Response to the Weakness 3**
>
> **W3:** **Applicability Concerns**: ... how the findings translate to **non-actor-critic methods** or across diverse benchmarks. For instance, assessing the proposed adaptive replay ratio method on a well-established sample-efficiency benchmark such as **Atari-100k, with Rainbow algorithm**, would improve the practical significance of the adaptive RR method.
>
> **A:** We thank the reviewer for the suggestion to extend our evaluations to **non-actor-critic methods and discrete control tasks**. This is an excellent complement to our previous experiments in continuous DMC tasks. We have completed additional experiments in the Atari-100k environment, utilizing DrQ($\epsilon$) as our baseline, which is a Rainbow DQN-based approach. Due to time and computational resource constraints, we have directly adopted the default replay ratio (RR=1) results from the [BBF](https://arxiv.org/abs/2305.19452) for all tasks, except for SpaceInvaders and Enduro. For the other experiments we conducted, each was performed with 5 random runs. In addition to experiments with three different static RRs, we also reran the ReDO experiments to serve as our baseline. The results are presented in the following table.
>
> 🌟 **Adaptive RR outperforms other configurations, including ReDO, in 11 out of the 17 tasks.** This not only demonstrates that Adaptive RR can strike a superior trade-off between reuse frequency and plasticity loss, but also shows that dynamically adjusting the RR based on the critic's overall plasticity level (specifically referring to the Q network in Atari) enables competitive performance through effective neuron-level network parameter resetting.
>
> |Task|RR=0.5|RR=2|ARR (from 0.5 to 2)|RR=1 (optimal static RR)|ReDO (RR=1)|
> |:---:|:---:|:---:|:---:|:---:|:---:|
> |Alien|815±133|917±132|**935±94**|865.2|794±47|
> |Amidar|114±58|133±57|**200±53**|137.8|163±56|
> |Assault|755±119|579±44|**823±94**|579.6|675±57|
> |Asterix|470±65|442±69|519±36|**763.6**|684±98|
> |BankHeist|451±114|91±34|**553±68**|232.9|61±43|
> |Boxing|16±9|6±2|**18±5**|9.0|9±5|
> |Breakout|17±6|13±3|16±5|**19.8**|15±6|
> |ChopperCommand|1073±318|1129±114|1544±519|844.6|**1650±343**|
> |CrazyClimber|18108±2336|17193±3883|22986±2265|21539.0|**24492±4641**|
> |DemonAttack|1993±678|1125±191|**2098±491**|1321.5|**2091±588**|
> |Enduro|128±45|138±36|200±32|**223.5**|**224±35**|
> |Freeway|21±4|20±3|**23±2**|20.3|19±11|
> |KungFuMaster|5342±4521|8423±4794|**12195±5211**|11467.4|11642±5459|
> |Pong|-16±3|**3±10**|-10±12|-9.1|-6±14|
> |RoadRunner|6478±5060|9430±3677|**12424±2826**|11211.4|8606±4341|
> |Seaquest|390±79|394±85|**451±69**|352.3|292±68|
> |SpaceInvaders|388±122|408±93|**493±93**|402.1|379±87|
>
> ---
>
> **We look forward to further discussions with the reviewer.**

---

> > ### Comment · Reviewer_7W3D · 2023-11-21
> >
> > Thank you for the detailed response and the additional experiments. The inclusion of various metrics in the supplementary section strengthens your paper, and **I've accordingly increased my score to 6**. While the expanded analysis using metrics like feature rank, active units, and weight norm is valuable, I would like to understand better how we can quantify the loss of plasticity, as these measures do not sufficiently quantify this loss.
> >
> > I encourage you to complete the pending experiments, as they are essential for a comprehensive understanding of the topic. Moreover, considering the quality and impact of the research, I believe this paper deserves a poster presentation at ICLR.

---

> > > ### Author Response · Authors · 2023-11-23
> > >
> > > Dear Reviewer 7W3D,
> > >
> > > Many thanks for your reply! We're glad that our response and new experiments address your concern.
> > > Your suggestions are crucial for enhancing the quality of our manuscript, and we will incorporate your recommendations into subsequent revisions.
> > >
> > > Thank you again for your precious time on the review and the appreciation on our work.
> > >
> > > Best regards,
> > >
> > > Authors of Paper4493

---

### Meta-Review · Area_Chair_dZvA · 2023-12-10

**Metareview:**

The work focuses on the issue of plasticity in visual RL and brings insight into the role of data augmentation in light of plasticity. The insight provided an adaptive way of addressing the replay ratio dilemma, which concerns the number of updates per interaction.

Strengths:

- The perspective of data augmentation in terms of plasticity is novel and insightful.

- The essentiality of critic plasticity is an insightful finding.

Weaknesses:

- Plasticity assessment relies on a limited metric.

- There has been confusion regarding whether the paper is a contribution to the understanding of plasticity in visual RL or the advancement of the state of visual RL. Most reviewers consider this paper as a significant contribution to the former but there is a disagreement regarding whether this work advances visual RL itself.

Decision:

As a work bringing a new understanding of plasticity in visual RL, I recommend acceptance of this paper. However, I strongly recommend the authors address the scope of the work and avoid making any claim that may mislead readers into thinking that the work brings advanced methods for visual RL.

The work also requires extensive discussions on many matters reviewers pointed out, such as the role between replay ratio and data augmentation, why a dramatically reduced fraction of active units in the actor was not an issue, and what specific mechanism of data augmentation helps with plasticity.

**Justification For Why Not Higher Score:**

To have a higher score, I would have expected the paper to provide a more substantial contribution, such as a method beyond the simple-heuristic-based one provided for the adaptive replay ratio or a clear contribution that is directly pertinent to and useful for the visual RL community.

**Justification For Why Not Lower Score:**

N/A

---

### Decision · Program_Chairs · 2024-01-16

Accept (poster)